# Behavioural individuality determines infection risk in clonal ant colonies

Zimai Li [1,2], Bhoomika Bhat [1], Erik T. Frank [3], Thalita Oliveira-Honorato[4], Fumika Azuma[5], Valérie Bachmann[2], Darren J. Parker [6], Thomas Schmitt [3], Evan P. Economo [5] & Yuko Ulrich [1,2,4] ✉

In social groups, infection risk is not distributed evenly across individuals. Individual behaviour is a key source of variation in infection risk, yet its effects are difficult to separate from other factors (e.g., age). Here, we combine epidemiological experiments with chemical, transcriptomic, and automated behavioural analyses in clonal ant colonies, where behavioural individuality emerges among identical workers. We find that: (1) *Caenorhabditis*-related nematodes parasitise ant heads and affect their survival and physiology, (2) differences in infection emerge from behavioural variation alone, and reflect spatially-organised division of labour, (3) infections affect colony social organisation by causing infected workers to stay in the nest. By disproportionately infecting some workers and shifting their spatial distribution, infections reduce division of labour and increase spatial overlap between hosts, which should facilitate parasite transmission. Thus, division of labour, a defining feature of societies, not only shapes infection risk and distribution but is also modulated by parasites.

Host spatial behaviour is a key driver of parasite infection and transmission success[1,2]. For example, host spatial distribution modulates infection risk, with feeding, breeding, or resting sites often acting as disease transmission hotspots[3,4]. Infections can in turn have drastic effects on host spatial behaviour, ranging from manipulations that increase parasite transmission between hosts to host behavioural responses that reduce transmission[5,6]. For example, some pathogenic fungi are thought to increase their transmission by causing infected ants to climb up and bite vegetation before dying so fungal spores are released from their cadaver onto foraging trails after their death[7]. Conversely, human and insect societies alike contain the spread of pathogens by reducing spatial overlap between group members[6,8]. Empirical studies on the relationship between host spatial behaviour and parasite infection risk largely rely on observations of natural populations[9–11], which are instrumental in generating hypotheses but can be confounded with other factors such as variation in age or

genotype[12,13]. Experimental tests of the link between host spatial behaviour and infection risk, however, are comparatively rare, in part because few systems allow controlled infection experiments and monitoring of individual behaviour in replicate host groups of known composition.

Social insect colonies are powerful systems to study the link between behaviour and infection because they display complex yet tractable spatial behaviour. While the core of the colony (the nest) is usually static, colony members exhibit heterogeneous spatial distribution around the nest that reflects their behavioural roles: for example, nurses mainly stay in the nest to care for the brood, while foragers leave the nest to collect food[14,15]. This spatial organisation of division of labour[14] has in turn been proposed to result in asymmetries in infection risk, with workers specialising in tasks outside the nest (e.g., foraging, waste management) expected to have increased exposure to parasites present in the external environment. While this

[1]Max Planck Institute for Chemical Ecology, Jena, Germany. [2]Institute of Integrative Biology, ETH Zürich, Zürich, Switzerland. [3]Department of Animal Ecology and Tropical Biology, Biocentre, University of Würzburg, Würzburg, Germany. [4]Department of Ecology and Evolution, University of Lausanne, Lausanne, Switzerland. [5]Biodiversity and Biocomplexity Unit, Okinawa Institute of Science and Technology Graduate University, Onna, Japan. [6]School of Natural Sciences, Bangor University, Bangor, UK. ✉e-mail: yulrich@ice.mpg.de

assumption is central to the study of disease transmission in social insect colonies[16,17], it is difficult to test rigorously. This is in part because behavioural roles in most social insect colonies covary with several individual traits, including age[14] and genotype[18]. These factors in turn typically affect disease resistance independently from behaviour[19–21], making it difficult to unambiguously measure the effects of behaviour on infection risk.

To overcome this problem, we investigated the link between behaviour and parasitic infection in the clonal raider ant (*Ooceraea biroi*). Colonies of this species are queenless and composed of genetically near-identical workers (within-colony relatedness is 0.996[22]) that reproduce asexually and synchronously[23], providing experimental control over genotype and age at the individual and group levels. Furthermore, colonies composed of identical individuals show spatially organised division of labour[24], i.e., individuality in spatial behaviour that reflects behavioural roles, making it possible to study the link between host behaviour and parasite infection in the absence of confounding variation in age or genotype. As an infectious agent, we used a nematode of the genus *Diploscapter*—the sister group of *Caenorhabditis*—that we find naturally infects *O. biroi* and other ants. Nematodes of this and related genera have sporadically been reported to infect ants since the late 19th century[25–30], but the nature of their interaction with ants and mode of transmission remains poorly characterised. The association between nematodes and ants displays high organ-specificity: nematodes were repeatedly reported to infect a specific head gland, the pharyngeal gland (PG)[25–29]. The PG has an eminently social function in ants, with roles in the transfer of nutrients and signalling molecules between colony members[31,32]. In particular, the PG plays a key role in the exchange of cuticular hydrocarbons (CHC) across colony members[33]. In ants, CHC profiles form the chemical basis for nestmate vs. non-nestmate discrimination, and are associated with behavioural roles[34]. Thus, while *Diploscapter* nematodes have often been assumed to use ants as a mere means of dispersal[28,29], the localisation of nematodes in the PG suggests that infections might affect the social organisation of the colony.

Here, we first confirm that *Diploscapter* nematodes naturally infect several ant species and localise to the PG. We also combine survival, transcriptomic, and chemical analyses of CHC profiles in experimentally infected clonal raider ants to show that these nematodes harm their host and broadly affect their physiology, indicating that they are not merely commensal. We then perform a long-term epidemiological experiment in colonies composed of identical workers to test whether workers active outside the nest face higher infection risk than otherwise identical workers active in the nest. Finally, we use experimental infections and automated tracking to test whether the presence of nematodes in the PG affects individual spatial behaviour and colony social organisation.

## Results

### *Diploscapter* naturally infects ants and affects host fitness and physiology

Sequencing confirmed that nematodes isolated from the heads of workers from *O. biroi* and two other ant species (*Paratrechina longicornis* and *Lasius niger*, Fig. 1a–b) cluster with *Diploscapter* sampled from the body and nest material of *Prolasius advenus* ants in New Zealand[30] (Fig. 1a). Thus, together with previous reports[26,28,29], our findings support the view that some *Diploscapter* nematodes naturally infect ants[35]. Micro-CT reconstructions of infected ant heads confirmed that *Diploscapter* localises to the PG in the studied species (Fig. 1b). Although the PG varies in shape and arrangement across species, in each case the nematodes fill the lumen of the gland, where they can reach high densities and become tightly packed in bundles within the different "fingers" of the glove-shaped PG. Infecting nematodes from the PG of *O. biroi* workers resemble dauer larvae (have a constricted pharynx; Fig. 1c, Supplementary Fig. 1), the developmentally arrested, non-reproducing larval stage that is also the infective stage in many parasitic nematodes[36].

While previous studies proposed that *Diploscapter* nematodes are commensals that use ants merely to disperse[28,29], our results instead show that these nematodes harm their host, and can therefore be (facultative) larval parasites. First, infections affected host survival. Experimentally infected clonal raider ant workers had a reduced survival rate (mean ± SD: 75.00 ± 4.84%) relative to otherwise identical uninfected workers (96.88 ± 1.78%) over 51 days (Cox proportional-hazards model: hazard ratio (95% confidence interval) = 9.96 (2.65, 37.47), z = 3.40, p = 0.0007; Fig. 2a). Second, infections strongly affected gene expression in the PG, including immune genes. We detected 657 differentially expressed genes in the PG (Supplementary Data 1) of clonal raider ants but no differentially expressed (DE) genes in an adjacent uninfected organ, the brain (Fig. 2b, Supplementary

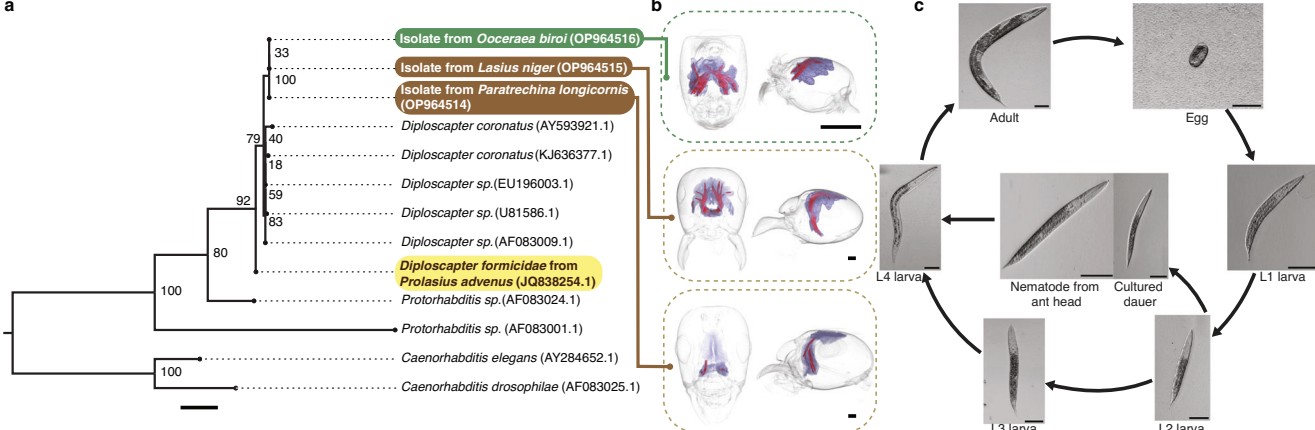

**Fig. 1 | *Diploscapter* nematodes infect the PG of ants. a** *Diploscapter* isolated from the heads of clonal raider ants (green, one isolate) and two other ant species (brown, one isolate per species) are phylogenetically nested within the *Diploscapter* clade (sequence accession numbers in brackets), which includes *D. formicidae* isolated from ants[30] (yellow). Node labels indicate branch support (%) from 1000 bootstrap replicates. Scale bar: mean number of 0.02 substitutions per site. **b** Micro-CT reconstructions confirm that nematodes (red) localise to the PG (purple) in all studied ant species (one specimen per species). Scale bars: 0.2 mm. **c** Life cycle of *Diploscapter* isolated from *O. biroi* (representative images from 5 replicate experiments). In favourable conditions, eggs hatch and go through four larval stages (L1–L4) before reaching the adult stage. If eggs hatch in unfavourable conditions (e.g., limited food), larvae develop into persistent, developmentally arrested dauer larvae. In favourable conditions, dauer larvae can resume development into L4 and adults. Scale bars: 50 μm.

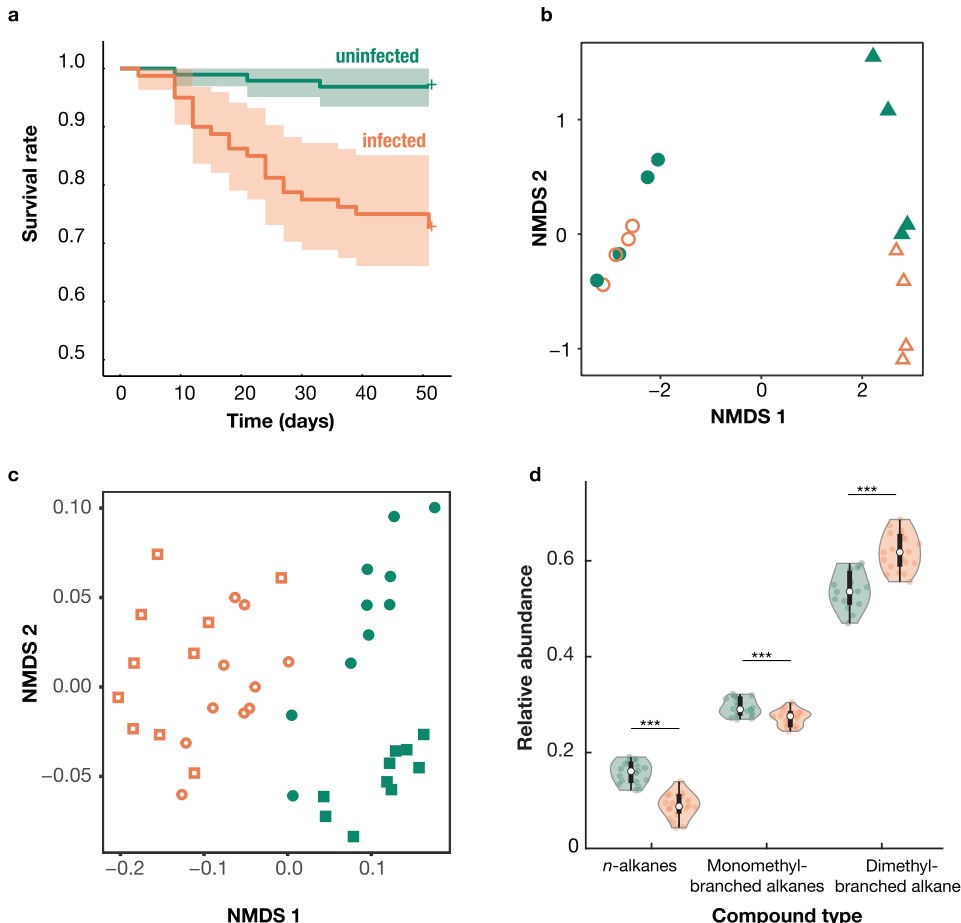

**Fig. 2 | Nematode infections affect host survival and physiology. a** Survival rate of infected ($n = 80$ workers in 10 colonies) and uninfected ($n = 96$ workers in 12 colonies) ants over time. Lines and shaded areas represent means and 95% confidence intervals. **b** Gene expression in the brain (circles) and PG (triangles) of infected (orange, open symbols) and uninfected (green, solid symbols) ants, visualised with a nonmetric multidimensional scaling (NMDS) analysis. Data points represent pooled samples of 5 brains or PGs. **c** CHC profiles of infected (orange, open symbols) and uninfected (green, solid symbols) ants from genetic lineages B (circles) and M (squares), visualised with an NMDS analysis. Data points represent pooled samples of 5 decapitated ants. **d** Relative abundance of CHC classes in infected (orange, right) and uninfected (green, left) ants. Thick black bars indicate the interquartile range (IQR) around the median (white circle), and whiskers represent 1.5 times the interquartile range (maxima: Q3 + 1.5 * IQR, minima: Q1 − 1.5 * IQR). Shaded areas are kernel density estimates. Data points represent pooled samples of 5 decapitated ants. Asterisks indicate the significance of chi-squared-test based model comparison. ***$p < 0.001$. Source data are provided as a Source Data file.

Data 2). Gene set enrichment analyses showed enrichment for a diverse set of gene ontology (GO) terms ($n = 333$), showing infection affected a broad range of functional processes. To aid interpretation we semantically clustered enriched GO terms and found 8 clusters with clear links to immune responses (e.g., defence response, response to bacterium, wound healing), as well as 9 clusters with clear links to behavioural processes (e.g., regulation of behaviour, social behaviour, aggressive behaviour) (Supplementary Data 3). The majority of DE genes in immune-related clusters showed increased expression in infected individuals (Supplementary Fig. 2, Supplementary Data 3), consistent with a modulation of the immune response. This effect was largest for genes related to wound healing (9/11 genes were upregulated) suggesting infection may cause physical damage to the PG. Lastly, infections affected host CHC profiles (ADONIS: $R^2 = 0.62$, $F = 76.79$, $p < 0.0001$; Fig. 2c, Supplementary Table 1). Our chemical analysis showed that infections altered the relative abundance of all CHC classes on the cuticle of clonal raider ant workers: both *n*-alkanes and monomethyl-branched alkanes had lower relative abundance in infected individuals (linear mixed models (LMM): infection status: *n*-alkanes: degrees of freedom (DF) = 1, likelihood ratio (LR) = 47.67, $p < 0.0001$; monomethyl-branched alkanes: DF = 1, LR = 15.20, $p < 0.0001$), while dimethyl-branched alkanes had greater relative

abundance (DF = 1, LR = 37.18, $p < 0.0001$) (Fig. 2d). Together, our findings show that some nematodes of the genus *Diploscapter* are facultative parasites of ants that affect the physiology and reduce the fitness of their hosts. Given the social function of the PG, we next investigated the link between nematode infection and the social organisation of the colony.

### Behavioural individuality determines infection risk

Like other nematodes, *Diploscapter* finds hosts by nictation[37,38], a stereotypical waving behaviour that allows attachment to passing hosts (Supplementary Movie 1). We hypothesised that hosts that differ in spatial behaviour (but are otherwise identical) may be at different risks of infection with such ambush-style parasites. We tested this hypothesis by exposing uninfected *O. biroi* colonies made up of age-matched, clonally related workers to agar seeded with nematodes over four colony cycles (216 days). Nictating nematodes attached to ants walking within their reach. As predicted, we find that infection risk is not homogenously distributed across colony members, but instead depends on individual behavioural roles. During the first colony cycle, nematode prevalence was already higher (mean ± SD: 90.00 ± 30.50%) in the workers acting as foragers outside the nest than in the otherwise identical workers acting as nurses in the nest (33.33 ± 47.90%)

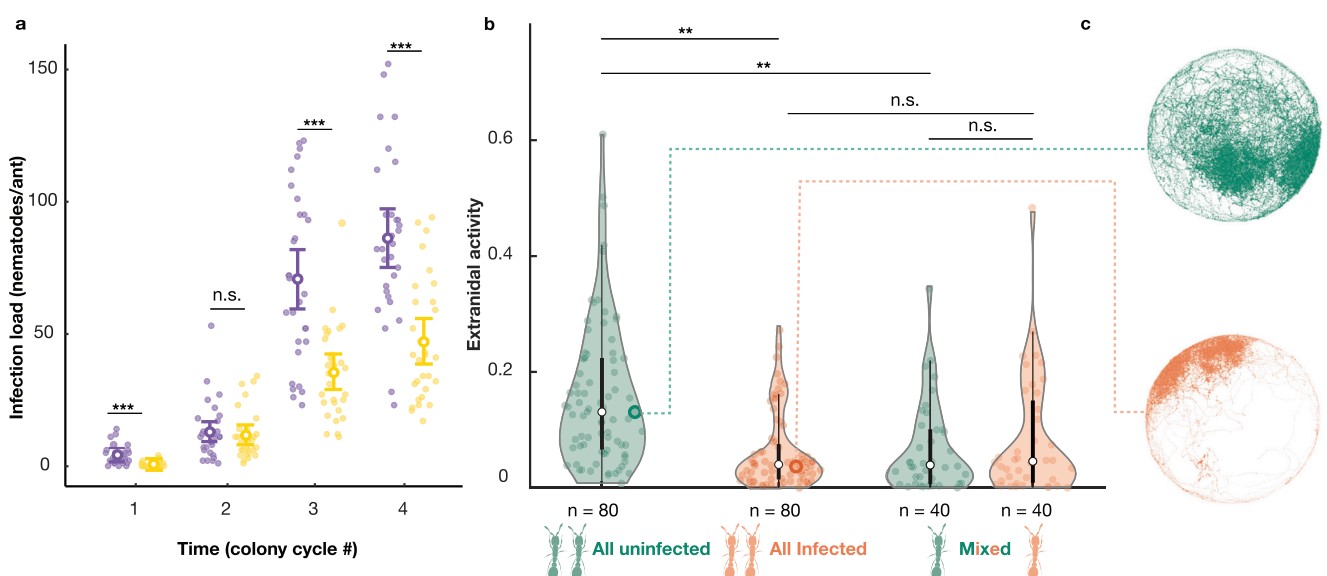

**Fig. 3 | Effects of behaviour on infection, and vice versa. a** Infection load in foragers (purple, left) and nurses (yellow, right) over time. Small opaque dots represent individual workers ($n = 30$ per behavioural role and colony cycle). Large circles and bars represent mean ± SD. Asterisks indicate the significance of differences in prevalence for colony cycle 1 (chi-squared test, $p < 0.0001$) and differences in infection load for colony cycles 2–4 (posthoc test with Tukey's adjustment for multiple comparisons; cycle 2: $p = 0.18$, cycle 3: $p < 0.0001$, cycle 4: $p < 0.0001$). **b** Extranidal activity (proportion of time outside the nest) of uninfected (green) and infected (orange) workers as a function of colony infection composition. Thick black bars indicate the IQR around the median (white circle), and whiskers represent 1.5 times the interquartile range (maxima: Q3 + 1.5 * IQR, minima: Q1 − 1.5 * IQR). Opaque dots and sample sizes represent individual ants. Shaded areas are kernel density estimates. Asterisks indicate the significance of posthoc tests with Sidak's adjustment for multiple comparisons ($Infected_{all}$ vs. $Uninfected_{all}$: $p = 0.008$, $Infected_{mixed}$ vs. $Uninfected_{mixed}$: $p = 0.983$, $Uninfected_{all}$ vs. $Uninfected_{mixed}$: $p = 0.008$, $Infected_{all}$ vs. $Infected_{mixed}$: $p = 0.999$). **c** Spatial distribution of representative workers (open circles in **b**) from an uninfected and an infected colony. ***$p < 0.001$, **$p < 0.01$, n.s. not significant. Source data are provided as a Source Data file.

(generalised linear mixed model (GLMM): worker behavioural role, DF = 1, LR = 22.19, $p < 0.0001$; Fig. 3a). In the subsequent cycles, all sampled ants were infected, but infection load increased faster in foragers than in nurses (GLMM: interaction between time and worker behavioural role, DF = 2, LR = 51.42, $p < 0.0001$; Fig. 3a), likely as a result of continued uptake from the environment. While in the second cycle, the mean infection load of foragers (12.80 ± 10.80 nematodes/ant) and nurses (11.60 ± 9.09) were undistinguishable (pairwise comparison of estimated marginal means (EMM): contrast ratio (95% confidence interval) = 1.10 (0.95, 1.28), z = 1.33, $p = 0.18$), in the third and fourth cycles, foragers had drastically higher infection loads than nurses (third cycle: foragers: 70.70 ± 32.20, nurses: 35.30 ± 17.40; 2.00 (1.86, 2.15), z = 18.44, $p < 0.0001$; fourth cycle: foragers: 86.40 ± 31.00, nurses: 46.90 ± 23.00; 1.83 (1.72, 1.96), z = 18.32, $p < 0.0001$). Thus, the distribution of parasites across hosts reflected inter-individual variation in host spatial behaviour driven by age- and genotype-independent division of labour.

**Infections alter social organisation**

Using automated tracking of experimentally infected hosts over 6 days, we find that not only does host spatial behaviour affect infection risk (Fig. 3a), but infection also affects host spatial behaviour (GLMM, DF = 3, LR = 11.24, $p = 0.01$; Fig. 3b–c). Workers in infected colonies spent more time in the nest (i.e., behaved more like nurses) than otherwise identical workers in uninfected colonies (extranidal activity: $Infected_{all}$: EMM (95% confidence interval): 0.06 (0.04, 0.09), $Uninfected_{all}$: 0.14 (0.10, 0.20); t = 3.14, $p = 0.008$). Unexpectedly, the presence of some infected individuals in the colony affected the behaviour of all colony members: the extranidal activity of uninfected workers in mixed colonies was as low as that of their infected nestmates ($Infected_{mixed}$: 0.06 (0.04, 0.10), $Uninfected_{mixed}$: 0.06 (0.04, 0.09), t = 0.46, $p = 0.983$)–and lower than that of uninfected ants in nematode-free colonies ($Uninfected_{all}$ vs. $Uninfected_{mixed}$, t = 3.14,

$p = 0.008$)–although they remained uninfected throughout the experiment (i.e., no transmission occurred, as confirmed by dissections). In contrast, the extranidal activity of infected ants was not increased by the presence of uninfected ants ($Infected_{all}$ vs. $Infected_{mixed}$: t = −0.14, $p = 0.999$). Individual infection load measured at the end of the experiment was not associated with individual extranidal activity measured over the first 6 days of the experiment (GLMM, LR = 0.025, $p = 0.875$, Supplementary Fig. 3). Because nematodes disproportionally infect foragers (Fig. 3a) and infections induce nurse-like spatial behaviour (Fig. 3b), our results imply that at the colony level, exposure to nematodes should effectively dampen division of labour and increase host density in the nest.

While we did not observe transmission in mixed colonies here, the initial infection load used was low (mean ± SD: 10.00 ± 4.16 nematodes/ant, $n = 4$ ants). To assess under what conditions horizontal transmission can occur, we performed additional mixed colonies experiments with higher initial infection loads (46.20 ± 22.40 nematodes/ant, $n = 10$ ants) at either low or high host density. We found that horizontal transmission readily occurs in these conditions (mean prevalence in originally uninfected ants: 73.86 ± 38.00%, $n = 19$ colonies) and that transmission increases at high host density (mean prevalence in originally uninfected ants: low density: 56.67 ± 45.44%, $n = 10$ colonies, high density: 92.96 ± 12.07%, $n = 9$ colonies; GLMM: DF = 1, LR = 5.61, $p = 0.02$; mean infection load in originally uninfected ants: low density: 1.63 ± 1.50 nematodes/ant, $n = 19$ ants, high density: 7.32 ± 8.12 nematodes/ant, $n = 80$ ants; GLMM: DF = 1, LR = 6.16, $p = 0.013$). By introducing uninfected ants into nest environments contaminated by infected ants (without direct contact between infected and uninfected ants), we also show that horizontal transmission can at least partially be environment mediated (mean prevalence in originally uninfected ants: 16.67 ± 14.43%, $n = 9$ colonies). Finally, we demonstrate that infecting nematodes can leave their live hosts, which likely facilitates horizontal transmission. Stereotypical nematode trails

($n$ = 4 plates) and a live nematode ($n$ = 1 plate) could readily be observed in the bacterial lawn of nematode culture plates ($n$ = 8 plates) 48 h after seeding these plates with one surface-sterilised ant each.

## Discussion

We report *Diploscapter* nematodes in the PG of three ant species and show that infections affect host fitness, physiology, and behaviour in *O. biroi*. The association of *Diploscapter* with ants is geographically widespread and organ-specific, but the nematodes retain the ability to complete their life cycle without hosts[29]. Our findings thus support the view that some *Diploscapter* nematodes are facultative larval parasites of ants[35] in a clade of otherwise free-living, bacterivore nematodes related to *C. elegans*[39]. Infections modulate immunity and decrease host survival, which together suggests that these nematodes do not merely use ants for dispersal, as previously thought[28,29], but may instead derive resources from their host. We suggest that the contents of the PG may make it an attractive source of nutrients for nematodes. The shift in host CHC profiles in infected workers, including reduced ratios of *n*-alkanes and monomethyl-branched alkanes, suggests nematodes may use hydrocarbons as a carbon source, potentially with a preference for certain compounds. In addition, the PG of ants contains sterols[40,41], which *C. elegans* (which belongs to the sister taxon of *Diploscapter*) requires exogenously for survival and to develop from dauer larvae into reproductive adults[42]. Further work is needed to establish what resources (if any) *Diploscapter* nematodes derive from ant PGs, and what role they play in their development.

Combining semi-natural and experimental infections with behavioural analyses, we show that *Diploscapter* nematodes interact with a defining feature of insect societies, division of labour. Workers active outside the nest became infected earlier and carried higher infection loads than otherwise identical individuals acting as nurses in the nest. Thus, differences in infection can result from differences in behaviour alone and the uneven distribution of parasites across group members can emerge from the spatial organisation of division of labour in the colony. This in turn highlights colony behavioural composition as a key driver of infection dynamics. Previous work in the clonal raider ant has shown that colony size[24], as well as colony demographic, genetic, and morphological composition[43] affect the spatial organisation of division of labour. Our results predict that each of these aspects of colony composition should in turn affect the dynamics of parasite acquisition in the colony. The optimal allocation of individuals to tasks in social insect colonies might therefore reflect a trade-off between the benefits (e.g., nutrient intake) and costs (e.g., parasite acquisition) of extranidal activity.

Not only did host spatial behaviour affect infection probability, but infections in turn affected host spatial behaviour at both the individual and colony levels. Experimentally infected workers spent considerably more time in the nest in proximity to the brood and nestmates. Surprisingly, the presence of infected ants also reduced the extranidal activity of their uninfected nestmates, potentially through social processes. Clonal raider ants forage in scout-initiated group raids, with one or a few scouts recruiting their nestmates to food outside the nest[44]. An infection-induced reduction of extranidal activity in some colony members would reduce the foraging activity of other colony members if it decreases the occurrence of scout-initiated raids. The observed reduction of extranidal activity upon infection contrasts with behavioural effects of infection with well-studied fungal pathogens, where infected hosts tend to reduce interactions with nestmates[8] or to leave the nest altogether[45–47]. Here, the infection-induced changes in host behaviour increased spatial overlap between hosts and are therefore expected to increase the nematode's transmission success to other colony members. From the parasite's perspective, gaining access to the nest of a social insect host is likely beneficial because, irrespective of the mode of transmission (contact-mediated vs. environment-mediated) and underlying causes of host

behavioural changes (e.g. sickness behaviour, parasite manipulation), the nest is invariably the place where host density and therefore transmission opportunities are highest.

Taken together, our behavioural analyses show that extranidal workers are more likely to become infected but that infections in turn reduce extranidal activity. Thus, by turning extranidal individuals into intranidal individuals, nematode infections effectively dampen division of labour at the colony level. Division of labour is thought to provide an organisational barrier against disease transmission in social insects because it creates spatial segregation between behavioural groups that can compartmentalise infections and slow disease spread[16,48–50]. By disrupting division of labour and increasing spatial overlap between colony members, infecting nematodes may collapse that organisational barrier to their own advantage.

## Methods

### Ants and nematodes

Colonies of the clonal raider ant are queenless and consist of genetically near-identical (within-colony relatedness is 0.996[22]), totipotent workers that reproduce asexually and synchronously, producing new workers in discrete cohorts[23,51]. This provides the ability to create replicate experimental colonies with precise control of genetic and demographic variation both within and between colonies. Synchronised reproduction drives stereotypical colony cycles lasting ca. 5 weeks[52], in which colonies alternate between reproductive and brood-care phases, corresponding to the absence and presence of larvae, respectively. During the reproductive phase, all ants remain in the nest and lay eggs. During the brood-care phase, the ants attend to the growing larvae at the nest but also leave the nest to forage, explore, or dispose of waste. Individual ants vary in their tendency to perform tasks in the nest vs. outside the nest, which is reflected in their spatial behaviour[24]. Colonies containing as few as 6 ants are fully functional and have division of labour (i.e. stable behavioural variation between individuals) under laboratory conditions in this species[24].

Nematodes were isolated from workers of laboratory colonies of *O. biroi* originally collected in Okinawa and the US Virgin Islands[53]. Because we cannot rule out transmission between colonies after collection, the exact geographical origin of the *O. biroi* nematode isolate was not known. To examine whether *Diploscapter* nematodes naturally infect other ants from different geographic locations, we dissected workers from other ant species that were available at the time of the study and obtained two further nematode isolates from laboratory-reared colonies of *P. longicornis* originally collected in Malaga, Spain, and *L. niger* originally collected in Lausanne, Switzerland.

Nematodes were isolated from dissected ant heads of each species and grown on agar plates seeded with bacteria (*Escherichia coli* OP50)[54]. To prepare infection inocula, nematodes were washed from agar plates with ddH$_2$O. The concentration of the resulting solution was determined (average of 3 independent counts per sample) in counting chambers (MultiCount10, Immune Systems Ltd., UK) and adjusted to the desired concentration (see below). Because infections are localised to the PG and live *Diploscapter* nematodes are not found elsewhere in the ant body, the infection status of individual ants can reliably be assessed by dissecting their head. Nematodes do not infect the brood, as shown by the fact that uninfected colonies can be established with newly-eclosed callow workers from infected colonies and remain uninfected for years. All experiments were performed with the *Diploscapter* isolate from *O. biroi* ants.

### Nematode phylogenetic analysis

We Sanger sequenced the SSU (18S rDNA) of three nematode isolates following Holterman et al.[55]. Sanger sequences were trimmed with Geneious Prime (v. 2021.0.3) to remove low quality bases using default options. Trimmed sequences were then assembled using the assembly tool in Geneious (with default options) to produce a consensus

sequence. We then downloaded all available *Diploscapter* 18 S sequences with >1000 bp of sequence from NCBI, as well as two *Protorhabditis* and two *Caenorhabditis* sequences, which are outgroups of *Diploscapter*[55]. Sequences were aligned using PRANK (v.100802, default options)[56]. The alignment was cleaned with Gblocks (v. 0.91, minimum block length = 50)[57] to eliminate poorly aligned positions. We then used RAxML (v. 8.2.12, options -m GTRGAMMA -p 12345 -# 1000 -f a -x 12345 -c 40 -T 10)[58] to generate a maximum-likelihood tree, assuming a GTR + gamma model of sequence evolution with 1000 bootstrap iterations.

## Micro-CT scans and segmentation

One infected worker of *L. niger*, *O. biroi*, and *P. longicornis* was scanned and each was given a unique specimen identifier (Supplementary Table 2). The specimens were fixed in 70% ethanol, stained in 2 M iodine solution for a minimum of 7 days, and then washed and individually sealed in a pipette tip with 99% ethanol for scanning. The scans were performed with the ZEISS Xradia 510 Versa 3D X-ray microscope at the Okinawa Institute of Science and Technology, Japan.

The head of each specimen was scanned at 40 kV and 3 W for a full 360° rotation with 1601 projections (except *O. biroi* which had 3201 projections). Exposure time was set to yield intensity levels between 13,000–15,000 across the head and ranged from 13–24.5 s. The resulting scans were reconstructed with the ZEISS Scout-and-Scan Control System Reconstructor software (ZEISS, Oberkochen, Germany).

The PG and parasites within the gland were manually segmented using the brush tool in Amira (v2019.2). The segmented materials were exported as Tiff image stacks, imported into VGStudio (v3.4; Volume Graphics GmbH, Heidelberg, Germany) and visualised using volume rendering (Phong).

## Nematode life cycle

To synchronise the nematode culture, eggs and gravid adults growing on one agar plate seeded with bacteria (*Escherichia coli* OP50)[54] were washed in 3 mL ddH$_2$O, centrifuged at 56 g for 1 min and the resulting pellet was resuspended in 3.5 mL of ddH$_2$O. The nematode suspension was vortexed in a mix of 0.5 mL 5 N NaOH and 1 mL 5% NaClO for 4 min to kill all nematodes except eggs and washed thrice by centrifugation and resuspension in 750 μL of ddH$_2$O to obtain a suspension of live eggs. 5 μL of the egg suspension was fixed on a glass slide in 2.5 μL of 0.25 mM levamisole for imaging. 50 μL each of the egg suspension was spotted on 4 fresh agar plates seeded with 100 μL of bacteria and incubated at 23 °C. Nematodes were picked from the plate by adding 10 μL of ddH$_2$O, fixed in 5 μL of 0.25 mM levamisole on a glass slide and imaged at ca. 7 h, 23 h, 41 h, 66 h, 91 h, and 115 h post-synchronisation, based on estimated times for transitions between successive developmental stages. A total of 5 nematodes were imaged for each developmental stage. To grow dauers, 400 μL of the egg suspension was spotted on an agar plate with 20 μL bacterial lawn and incubated at 23 °C. To image dauers, all nematodes on the agar plate were washed in 3 mL ddH$_2$O after ca. 115 h post-synchronisation and centrifuged at 56 g for 1 min to obtain a pellet. The resulting pellet was resuspended in 2 mL 1% SDS solution and incubated with shaking at 180 rpm for 15 min at room temperature to kill all nematodes except dauers. SDS was then removed by centrifugation at 56 g and washing the pellet in ddH$_2$O to obtain a dauer suspension. 10 μL of the dauer suspension was fixed in 5 μL of 0.25 mM levamisole and a total of 5 dauers were imaged. To image infective stage nematodes, the heads of 5 *O. biroi* workers were dissected in ddH$_2$O and 5 nematodes from each ant were mounted on a glass slide in 10 μL ddH$_2$O and fixed with 5 μL of 0.25 mM levamisole. All images were captured with a Zeiss AXIO Zoom.V16 microscope at a final magnification of 412.1x in the Relief Contrast mode, using the software ZEN Blue.

## Survival assay

Uninfected, age-matched (77-days old), clonally related (genotype A) workers were divided into two groups: infected (140 ants) and uninfected (140 ants). Each group was kept in a plastic nest box (110 × 150 × 80 mm) with a plaster of Paris floor. Ants from the uninfected group were kept nematode-free, while ants from the infected group were exposed to nematodes by the addition of 10–100 infected ant heads in their nest box every 4–11 days. 75 days post-exposure, the success of the treatment was confirmed by dissecting 10 ants per treatment: all ants from the uninfected group were nematode-free and all ants from the infected group carried nematodes (6.8 ± 1.29 nematodes/ant). Ants were split into 10 infected and 12 uninfected experimental colonies in airtight Petri dishes (diameter 50 mm) with a plaster of Paris floor. Each experimental colony consisted of 8 ants and 6 larvae. Every 3 days over 51 days, colonies were watered and fed with frozen ant (*Messor*) brood, worker survival rate was monitored, and any dead ants were replaced with genotype- and age-matched ants from the corresponding treatment (uninfected or infected) to rule out effects of declining group size on survival rate.

## Transcriptomic analysis

Brains and PGs were dissected from 20 infected and 20 uninfected age-matched (85-days old), clonally related (genotype A) workers. The workers had been exposed to nematodes on agar (infected) or nematode-free agar (uninfected) as a group for 55 days prior to dissection. Brains or PGs of 5 workers were pooled to produce 4 samples per tissue type. RNA was extracted from each sample using a modified Trizol/phenol chloroform protocol following Libbrecht et al.[59]. Libraries were prepared using a KAPA stranded mRNA-sequencing kit following the manufacturer's protocol. Libraries were sequenced using standard protocols on an Illumina HiSeq 4000. Raw sequencing reads for infected and uninfected brains and PGs have been deposited in NCBI's sequence read archive under the following BioProject accession: PRJNA791185 .

Reads were trimmed using Trimmomatic (v. 0.36)[60], to remove adaptors and low-quality bases (LEADING:9 TRAILING:9 SLIDINGWINDOW:4:15). Reads less than 90 bp after trimming were discarded. Trimmed reads were then mapped to transcripts extracted from the *O. biroi* reference genome (GCF_003672135.1_Obir_v5.4[61]) with Kallisto (v. 0.46.0)[62]. Read counts for each gene were then obtained using *tximport* (v. 1.10.1) in R (v4.0.3)[63] (Supplementary Table 3). Expression analyses were performed using the Bioconductor package *EdgeR* (v. 3.24.0)[64]. rRNA genes and genes with counts per million <0.5 in two or more libraries per condition were excluded. Expression analyses were performed separately for each tissue. Normalisation factors for each library were computed using the TMM method. To estimate dispersion, we fit a generalised linear model with a negative binomial distribution with infection status as an explanatory variable. A quasi-F test was used to determine the significance of infection status from this model for each gene, with p values corrected for multiple tests using Benjamini and Hochberg's algorithm. Statistical significance was set to 5%.

Genes were functionally annotated using Blast2GO[65] as follows: Gene sequences were compared with blastX to either NCBI's nr-arthropod or *Drosophila melanogaster* databases, keeping the top 20 hits with e-values < 1 × 10$^{-03}$. Interproscan (default settings within Blast2GO) was then run for each sequence, and the results merged with the BLAST results to obtain GO terms. This produced two sets of functional annotations, one derived from all arthropods and one specifically from *D. melanogaster*. The *D. melanogaster* GO term annotation generated around four times more annotations than NCBI's nr-arthropod database. We therefore conducted all subsequent analyses using the GO terms derived from *D. melanogaster* but note that enrichment analyses using the annotations from all arthropods found broadly similar terms albeit with fewer and less specific annotations.

Overrepresented GO terms were identified by conducting gene set enrichment analyses (GSEA) using the R package *TopGO* (v. 2.28.0), using the elim algorithm to account for the GO topology. GO terms were considered to be significantly enriched when $p < 0.05$. Enriched GO terms were then semantically clustered using ReviGO[66] to aid interpretation.

## Chemical analysis

To quantify differences in CHC profiles between infected and uninfected ants, we sampled ants from infected and uninfected colonies of two genotypes (B and M; 50 workers for each genotype and infection status combination). All colonies were in the reproductive phase, ruling out potentially confounding effects of variation in colony phase on CHC profiles (ants vary in behavioural, nutritional, and reproductive status across colony phases). Heads were dissected to confirm infection status and headless bodies were pooled in groups of 5, producing 10 samples per genotype-infection status combination for further analyses.

CHCs were extracted from pooled samples by immersion in 1 mL of hexane for 10 min. Extracts were then evaporated to a volume of approximately 15 μL of which 1 μL was analysed using a 6890 gas chromatograph (GC) coupled to a 5975 mass selective detector (MS) (Agilent Technologies, Waldbronn, Germany). The GC was equipped with a DB-5 capillary column (0.25 mm ID × 30 m; film thickness 0.25 μm, J & W Scientific, Folsom, CA, USA). Helium was used as a carrier gas with a constant flow of 1 mL/min. A temperature program from 60 °C to 300 °C with 5 °C/min and finally 10 min at 300 °C was employed. Mass spectra were recorded in the EI mode with an ionisation voltage of 70 eV and a source temperature of 230 °C. The software ChemStation v. F.01.03.2357 (Agilent Technologies) for Windows was used for data analysis. Identification of the compounds was accomplished by comparison of library data (NIST 17) with mass spectral data of commercially purchased standards for *n*-alkanes, diagnostic ions and retention indices. To calculate the relative abundance of CHC compounds, the area under each hydrocarbon peak in the chromatogram was quantified through integration and divided by the total area under all hydrocarbon peaks.

## Effects of behaviour on infection

Uninfected, age-matched (74-days old), clonally related (genotype A) ants were divided into three infected colonies (120 ants each) and one control colony (50 ants). The control colony was used to verify that the ants remained nematode-free during the whole period of the study and was not included in statistical analyses. All groups were kept in nest boxes (110 × 150 × 80 mm) with a plaster of Paris floor. For each infected colony, 1.5 g of agar was placed in the box and seeded once with 100 μL of a nematode inoculum (concentration: 189.60 nematodes/μL). For the control group, nematode-free agar (1.5 g) was used.

In each colony, ants were dissected at 4 time points corresponding to the brood care phase of 4 successive colony cycles. The first time point was 35 days after the start of the experiment for all colonies. After this, cycles became slightly desynchronized across colonies and for each infected colony, the subsequent dissection occurred 8 days after the eclosion of new workers, corresponding to the middle of the brood care phase, when foraging activity is at its peak. At each time point, the heads of 10 workers collected in the nest ("nurses") and 10 workers collected outside the nest ("foragers") were dissected under a stereomicroscope and infection load was determined by counting nematodes. Colony size was kept constant by removing all newly eclosed callows at each colony cycle, except for 20 callows that were paint-marked (to be distinguished from the original workers) and returned to the colony to replace the 20 dissected ants. Workers from the uninfected control colony were dissected at each time point to ensure that no contamination took place. The experiment lasted 216 days in total.

## Effects of infection on behaviour

Uninfected, age-matched (20-day old), clonally related (genotype A) ants were divided into 2 groups: infected (150 ants) and uninfected (150 ants). Each group was kept in a plastic nest box (110 × 150 × 80 mm) with a plaster of Paris floor. Ants from the uninfected group were kept nematode-free, while ants from the infected group were exposed to 100 μL of a nematode inoculum (concentration: 280 nematodes/μL) seeded onto 1.5 g of agar. 45 days post-exposure, 4 ants from the infected group were dissected and confirmed to be infected (mean ± SD: 10.00 ± 4.16 nematodes/ant, range: 5–15 nematodes /ant). Ants were split into 30 experimental colonies in airtight Petri dishes (diameter 50 mm) with a plaster of Paris floor: 10 uninfected colonies, 10 infected colonies, and 10 mixed colonies (with half infected and half uninfected workers). Each experimental colony consisted of 8 workers (by that time 65-day old) and six 6-day-old larvae. Workers were tagged with unique combinations of paint marks (Uni Paint PX-20 and PX-21) on the thorax and gaster to be individually detected in video analyses.

Behavioural data were acquired from all experimental colonies over the first 6 days of the experiment. During this period, all colonies were in the brood care phase and no ant died in either treatment. 20 min of video (10 frames per second) were recorded every 2 h throughout this period using webcams (Logitech C910). Every 3 days until the end of the experiment, the colonies were watered and fed live ant brood (*Messor*), and any dead ants were removed. For each colony, the experiment ended when all larvae had eclosed into new adults (i.e., when the colony completed a cycle, which took on average 33.96 ± 7.06 days, range: 30–54 days) or after 60 days (if the colony failed to complete a cycle). At the end of the experiment, the heads of all workers in each colony were dissected and nematodes were counted.

Individual trajectories were extracted from videos using the software anTraX[67]. Extranidal activity, defined as the fraction of time an ant was outside the nest, was computed using MATLAB v.2022a (MathWorks, Natick, MA, USA) following Jud et al.[52].

## Additional experiment on transmission

To assess the conditions under which horizontal transmission occurs, we created mixed colonies consisting of infected ants (genotype A, mixed ages, mean infection load: 46.20 ± 22.40 nematodes/ant, range: 20–87 nematodes/ant, $n = 10$ ants) and uninfected ants (genotype B, 87 days old) in equal proportions. These colonies were established at either low density (8 ants per colony, $n = 10$ colonies) or high density (20 ants per colony, $n = 9$ colonies) in airtight plastic containers of the same size (diameter 33 mm) with a plaster of Paris floor. Infected ants were paint-marked to indicate infection status. Additionally, two control colonies of 10 uninfected ants each were used to verify that ants remained nematode-free during the experimental period and were not included in statistical analyses. Colonies were watered and fed frozen ant brood (*Tetramorium* sp.) three times per week (every other weekday). During the first 31 days, when an ant died, its cadaver was left in the colony, and a new tagged ant with the same infection status was added to maintain group size constant. After 31 days, we dissected the heads of all originally uninfected ants in each colony and counted nematodes. After 59 days, all remaining infected ants and cadavers were removed and 6 uninfected ants (genotype B, 31 days old) were added to each of the (now putatively nematode-contaminated) plastic containers in the high density treatment. After 73 days, we dissected all originally uninfected ants in each colony and counted nematodes. To assess the ability of parasites to leave their live hosts, we placed one surface-sterilised live ant onto nematode culture plates (60 mm diameter agar plates seeded with *E. coli* OP50; $n = 8$ plates) and used a

stereomicroscope to screen plates for nematodes and nematodes trails after 48 h.

## Statistical analyses

All statistical analyses were conducted in R 4.1.2[63].

The effect of treatment (infected vs. uninfected) on individual survival was assessed using a Cox proportional-hazards mixed model (function *coxme* from package *coxme*[68]) with colony as a random effect. Model assumptions were verified using the *cox.zph* function from the package *survival*.

CHC profiles were visualised with a nonmetric multi-dimensional scaling (NMDS) analysis using a Bray-Curtis dissimilarity matrix with the *metaMDS* function from the package *vegan*[69]. The effect of infection status (infected vs. uninfected) on CHC profiles composition was analysed using a permutational multivariate analysis of variance (ADONIS) on the Bray-Curtis dissimilarity matrix with genotype as a random effect with the *adonis* function from the package *vegan*[69]. To model the relative abundance of each of the CHC classes (*n*-alkanes, monomethyl-branched alkanes, dimethyl-branched alkanes), we used a LMM (function *lmer* from package *lme4*) with infection status (uninfected vs. infected) as a fixed predictor and genotype as a random effect.

To model nematode prevalence in the first colony cycle, we used a binomial GLMM with logit link function (function *glmer* from package *lme4*) with individual infection status (infected vs. uninfected) as response variable, worker behavioural role (forager vs. nurse) as fixed predictor, and colony as a random effect. To model infection load in the following cycles, we used a Poisson GLMM with log link function (function *glmer* from package *lme4*) with colony cycle (a three-level factor), worker behavioural role (forager vs. nurse), and their interaction as fixed predictors, and colony as a random effect. Pairwise comparisons between the estimated marginal mean (EMM) of infection load of foragers and nurses at colony cycles 2 to 4 were performed using posthoc tests implemented with the *emmeans* function from the package *emmeans*. P values were adjusted using the Tukey method for multiple comparisons.

To model worker extranidal activity, a beta GLMM with logit link function was implemented with the function *glmmTMB* from the package *glmmTMB*[70]. The model included a four-level variable combining infection status and colony composition (Uninfected$_{all}$, Infected$_{all}$, Uninfected$_{mixed}$ and Infected$_{mixed}$) as fixed predictor and colony as random effect. Pairwise comparisons between the EMM of extranidal activity between selected groups (Uninfected$_{all}$ vs. Infected$_{all}$, Uninfected$_{mixed}$ vs. Infected$_{mixed}$, Uninfected$_{all}$ vs. Uninfected$_{mixed}$, Infected$_{all}$ vs. Infected$_{mixed}$) were performed using posthoc tests implemented with the *contrast* function from package *emmeans*. *P* values were adjusted using the Sidak method for multiple comparisons. To test the association between worker extranidal activity (measured over experimental days 1–6) and infection load (measured at the end of the experiment, i.e., between days 30 and 60, depending on the colony), we used a LMM (function *lmer* from package *lme4*) with extranidal activity as response variable, infection load as a fixed predictor, and colony as a random effect.

To model nematode prevalence in colonies with different host densities, we used a binomial GLMM with logit link function (function *glmer* from package *lme4*) with individual infection status (infected vs. uninfected) as response variable, host density (low vs. high) as fixed predictor, and colony as a random effect. To model nematode infection load in colonies with different host densities, we used a Poisson GLMM with log link function (function *glmer* from package *lme4*) with individual infection status (infected vs. uninfected) as response variable, host density (low vs. high) as fixed predictor, and colony as a random effect.

LMMs and GLMMs models were compared to reduced models using likelihood ratio tests to assess the significance of predictors using the function *drop1* in R. Assumptions of LMMs and GLMMs were verified using model diagnostic tests and plots implemented in the package *DHARMa*.

## Reporting summary

Further information on research design is available in the Nature Portfolio Reporting Summary linked to this article.

## Data availability

The behavioural and chemical data generated in this study have been deposited in an Edmond repository (https://doi.org/10.17617/3.I6NKXM). The RNAseq data have been deposited at NCBI's sequence read archive under the following BioProject accession: PRJNA791185. Accession numbers for nematode sequences from this study are: isolate from *O. biroi* (OP964516), isolate from *L. niger* (OP964515), isolate from *P. longicornis* (OP964514). Accession numbers for nematode sequences from previous studies are: *Diploscapter coronatus* (AY593921.1), *Diploscapter coronatus* (KJ636377.1), *Diploscapter* sp. (EU196003.1), *Diploscapter* sp. (U81586.1), *Diploscapter* sp. (AF083009.1), *Diploscapter formicidae* from *P. advenus* (JQ838254.1), *Protorhabditis* sp. (AF083024.1), *Protorhabditis* sp. (AF083001.1), *Caenorhabditis elegans* (AY284652.1), *Caenorhabditis drosophilae* (AF083025.1). Source data are provided with this paper.

## Code availability

All R scripts for statistical analyses are available in the Edmond repository (https://doi.org/10.17617/3.I6NKXM).

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

## Acknowledgements

The authors thank Martin Niebergall, Giacomo Alciatore and Stephanie Jud for assistance with automated tracking, Daniel Knebel, Alexandre Courtiol, and Grit Kunert for advice on data analysis, Hugo Darras and Jason Buser for sharing ant samples, OIST Imaging Section for access to the Micro-CT Scanner, Christine La Mendola for assistance with RNA library preparation, Philipp Schiffer and the Gönczy lab (in particular Alexandra Bezler) for teaching us basic nematology, Veit Grabe for microscopy training, Oleksandr Holovachov for help on nematode life stage identification, Daniel Veit, Tim Zetzsche, and Qi Wang for assistance in setting up experiments, and members of the Social Behaviour group for comments on the manuscript. Sequencing was performed at the Genomics Technologies Facility of the University of Lausanne. This work was supported by the Max Planck Society, the Swiss National Science Foundation (grant no. PCEFP3_187005), and the European Research Council (ERC) under the European Union's Horizon 2020 research and innovation programme (grant agreement no. 851523) to Y.U.; E.F. was funded by the German Research Foundation (DFG, project no. 511474012). This is Clonal Raider Ant Paper #30.

## Author contributions

Z.L.: sample collection, data acquisition, data curation, data analysis, writing—original draft, writing—review and editing; B.B.: data acquisition, data analysis (nematode life cycle), writing—review and editing; E.F.: data acquisition, data analysis (chemistry), writing—review and editing; F.A.: data acquisition, data analysis (micro-CT); V.B.: Data acquisition; T.OH.: data acquisition, data analysis (behaviour); D.J.P.: data analysis (sequencing), writing—review and editing; T.S.: data analysis (chemistry), writing—review and editing; E.E.: data acquisition, data analysis (micro-CT), writing—review and editing; Y.U.: conceptualisation, funding acquisition, methodology, sample collection, project administration, resources, supervision, validation, writing—original draft, writing—review and editing. All authors contributed, reviewed and approved the manuscript for publication.

## Funding

## Competing interests

The authors declare no competing interests.
