## [Peer Review File · Nature Communications]

Behavioural individuality determines infection risk in clonal ant coloniesReviewers' Comments:

Reviewer #1:

Remarks to the Author:

The authors present a fascinating study and demonstrate that behavioral individuality causes differences in infection. Most notably the authors use a tractable and unique animal system -clonal raider ants- which allows them to disentangle the effects of other factors (e.g., genotype) on infection load from behavioral variability alone. They also demonstrate that the infection can affect social organization because infected workers stay in the nest. While I find this an interesting study, I have some concerns that I would suggest being addressed before I can recommend publication.

General comments:

Both the title and the abstract suggest that this is a study that predominately focuses on the behavior of infected ants. But, there are parts of the manuscript which appear somewhat disconnected in its current form (and are not mentioned in the abstract/intro). For instance, the authors have a whole section on how they test for natural infection in different species, how infection changes the ants' physiology (survival and potentially fitness, including gene-expression work, CHC abundance), and where the infection is localized in the brain (Micro-CT), but the connection of these experiments to the behavioral component of the work is often a bit unclear (i.e., why were all these additional experiments included?). I, therefore, suggest clarifying this connection between the different, successive steps of the authors' experimental work. This could be achieved in several sections such as the abstract, introduction, and results (see more specific comments below).

While the link to the data repository works, the provided DOI (<https://doi.org/10.17617/3.I6NKXM>) for the repository does not seem to exist. I suggest correcting this.

Abstract

I think the abstract addresses all the behavioral experiments clearly (and their background) but there is a whole set of experiments (e.g., PG infections, gene expression comparisons, Micro-CT, tests for natural infection) that are not mentioned which, to me, leads to a disconnect and questions of why these additional experiments were included. I think the authors could relatively easily incorporate why all these experiments were done and how they connect to the main experiment that investigated behavioral individuality and its effect on infection load (see comment above).

Minor comment

Line 17: The authors mention other factors. I believe that this is a key sentence that distinguishes their work from other work (and highlights how unique their study system is). I would therefore suggest mentioning such other factors that could lead to variation in infection risk in the abstract.

Introduction

Overall, the introduction is well-written. I do feel like it ends abruptly without a quick summary of what was done and what was predicted by the authors. More specifically the introduction ends with a statement about how the localization of the nematodes might affect social organization in the colony. I believe this would be an opportunity to briefly follow up connect the authors' experimental approaches (see general comment above) by stating that they first confirmed natural infection, localization of the infection, and changes to physiology (using different methods), and then used this parasite to investigate its effects on social behaviors and downstream social organization. This statement could also include some predictions (i.e., we predicted the parasite to be localized in the PG and therefore it could affect social behaviors). I believe such a summarizing statement at the end of the introduction

could clarify some of my main concerns about the disconnect of physiological and behavioral work.

I think the introduction could benefit from a bit more details about the transmission dynamics of the nematodes (if known). Namely, (1) how do ants get infected, (2) does the parasite spread horizontally? Is increased infection load a function of replication or continued uptake from the environment? What is the life cycle, and how long from infection to shedding? These add-ons could be made in the paragraph that discusses the biology of the parasite (lines 63-75).

Minor comments:

Lines 37-38: I suggest mentioning these other factors more explicitly since they are important to differentiate the authors' study from other studies in this field.

Line 55: Here and throughout I suggest changing to past tense. I.e., "investigated" instead of "investigate".

Line 58: This is a bit confusing. If they are near-identical can the authors exclude genotypic effects? Please clarify.

Line 67: I suggest either removing "Strikingly" or adding a clarification on why this specific tissue tropism is striking compared to other infection types.

Lines 68-69: I would suggest removing the statement about the previous name of the PG. I don't think it is necessary information and might confuse the reader.

Results

The results are well presented and I very much like the presented figures.

General comments:

For some of the visualizations, I wonder whether it would be worth it to make them more accessible for non-color printing. I.e., instead of using colors also use different symbols. But, I leave this up to the editors and authors discretion.

Minor comments:

Line 94: I suggest defining "phoretic", i.e., attachment for the purpose of travel here.

Line 94: The authors state that their results indicate that the nematodes do not use their ants host for dispersal alone and that this could be driven by reduced survival of the infected workers over 51 days. From the figures, it looks like survival differences started to arise around ~10 days. Does this mean that parasites do not disperse in these early days of infection? I suggest clarification about some of the potential transmission dynamics around the presented results (e.g., in the introduction, see comments above).

Line 115-116: I think this sentence is a great opportunity to establish the connection of why the physiological experiments were done and how they connect to the behavioral work. This could be achieved by, for instance, stating "because we also detected changes in CHC classes on the cuticle important for social interactions in this species, we next tested...."

Line 138-140 This is interesting, so nurses, despite being predominantly in the nest, also get infected, and infection load increases? Does this mean there is horizontal transmission? Again, I think a

description of the transmission dynamics (in the intro) could clarify some of these points.

Discussion:

General comment:

This is a general thought and might be something the authors wish to discuss. At the very end of the discussion, the authors mention that "by disrupting division of labor and increasing spatial overlap between colony members, infecting nematodes may collapse the organizational barrier to their own advantage". But, couldn't it also be detrimental to the parasite if the infection stays localized within the nest and it does not disperse beyond its barriers?

Minor comments:

Lines 179-181: I am a bit confused about how the authors results indicated that this specific parasite represents an early, incomplete transition to parasitism. Please clarify.

Lines 183-185: Does immune activation always mean that the intruder derives resources from the host? Please clarify.

Lines 188-189: The statement about *C.elegans* seems disconnected. I suggest clarifying the relationship of *C.elegans* with *Diploscapter* here (sister group) to establish the connection that sterols could also affect *Diploscapter*.

Lines 216-218: I think this statement could again benefit from more clarification about the transmission dynamics (see comments about the intro above). Is there horizontal transmission?

Lines 224-225: This sentence suggests that the infected workers just continue their "work" inside the nest which I assume isn't the case since they are probably suffering from the heavy effects of the infection. I wonder if it would make more sense to replace "workers" with "individuals" (in this sentence at least).

Methods:

The methods are well-described and easy to follow. I also believe they provide enough detail for the work to be reproduced.

Minor comments:

Lines 240-241: The sentence read to me that the same ants tend to larvae but also leave the nest to forage during the brood-care phase, but I assume this is when reproductive division of labor occurs and some ants are nurses and others are workers, correct? Maybe slightly rewrite to clarify this?

Reviewer #2:

Remarks to the Author:

Li and co-authors conducted a series of experiments involving various approaches that include molecular phylogeny, micro-CT, transcriptome, CHC analysis and behavioral tracking to show that behavioral variations (division of labor) by the clonal raider ant is a key determinant of infection risk posed by a parasitic nematode. They also showed that nematode infection modulates the ant's social organization in which individuals within infected colonies overlap spatially which in turn may favor the

nematode transmission.

All the experiments and analyses were well carried out. Data presentation is clear and straightforward. I particularly appreciate the micro-CT efforts that produce the very first empirical evidence for where these parasitic nematodes are located in an ant head (without dissection). Furthermore, I am happy to see that nematodes are now added on the list for the study of interactions between sociality/social behavior and disease dynamics. In general, authors have done a terrific job addressing the key questions with solid data, and most of the conclusions are well supported by the results.

One critical question that may need a clarification is that are nematodes able to transmit between ant individuals and, if they are, what would be the primary transmission pathway and how efficient is it? As authors mentioned, these nematodes are using nictation behavior to attach on a host ant that passes by and then enter the ant's PG. Do they have to leave the ant host's PG so they can be picked up by another ant worker? Authors must establish this baseline information to support their prediction on spatial overlapping of individuals facilitating parasite transmission. Also, authors may want to look at and test if trophallaxis serves as an effective/promising transmission route for the nematode to spread among nestmates, especially given the fact that the nematodes are localized in PG that plays some significant roles when it comes to sharing food via trophallaxis (many ants, not sure about *Ooceraea biroi* though, mix food with PG secretion before regurgitation)? If it is true that trophallaxis serves as a major way for nematodes to spread, just workers staying close with each other may not necessarily result into increase chance of transmission unless trophallaxis frequency can be established. Another potential scenario would be "sickness behavior" - reduced extranidal activity may simply be an adaptive behavioral change (e.g., ants conserve energetic resources so they are able to sustaining nematode infection) responding to nematode infection. The authors may also want to account this possibility when interpreting the results.

I also identified a few places that may use a nice justification to make the statements clearer:

L347: what are the potential confounding factors that may influence the CHC profiles of the ant during colony phase

L389: I wonder if infection load was quantified in this experiment? I understand that authors did present the nematode load data in the section "Behavioural individuality determines infection risk", but I was not sure if the nematode load data were presented in the section "Infections alter social organisation"? The reason I would like to have authors address this question is that it is likely that those infected individuals that still engaged extranidal activities be those who have a lower nematode titer (counts)?

L402-403: I appreciate the efforts that authors dissected all workers at the end of experiments. I wonder how many individuals from the infected groups actually have no nematodes in their PGs? I asked this question because this would provide readers information regarding how efficient/successful the artificial inoculation assay is (for example, does it result into 100% infection rate among workers in a group?).

Nematode movement in general: My personal experiences and also previous literature (e.g., Poinar, 2012, *Psyche*, 192017) suggest that these nematodes enter their host ants when the external conditions are not favorable (such as low soil humidity) and may leave the host ants when conditions are favorable again. Have the authors observed nematodes exit the ants? I assume humidity was well controlled in these experimental units, which may have created an environment with suitable conditions for the nematodes. I guess the answer to the question above (L402-403) will also address this issue. ^[1]_{SEP}

The introduction section would use a nice final paragraph that summarizes what hypothesis will be tested and what tools will be used for testing these hypotheses.

Reviewer #3:

Remarks to the Author:

This study by Li et al. presents an exciting behavioral, chemical, and genetic investigation of nematode infection in the clonal raider ant, *Ooceraea biroi*. I find this research to be of great interest for its contribution to our understanding of the effects of *Diploscapter* infection on ant behavior (specifically, forager extranidal behavior in *O. biroi*) and well as the role of forager ant behavior in infection likelihood. The authors also present interesting results regarding the effects of nematode infection on ant gene expression – particularly as it relates to immune response and behaviorally-associated genes, as well as results on changes to *O. biroi* CHC profiles, which are important for social group cohesion and non-nestmate discrimination, among other roles. In general, the authors do an excellent job of summarizing their findings and discussing the implications of their work and I think this study opens a lot of interesting questions about what might be happening behaviorally inside the colony during infection, how transmission occurs, and whether ants might be able to detect infected individuals through changes in CHCs. While most of my detailed comments in the manuscript relate to minor suggested edits and/or clarifications, a few comments relate to some of the conclusions drawn (summarized below). I have also uploaded my detailed, in-text comments included as attachments for both the manuscript and supplemental figures.

Comments/suggestions (by line number):

Main manuscript:

59: Missing a comma after i.e.

81: Highlight the *P. advenus* sample in Figure 1a to make it easier for readers to identify

90: Relating to the previous comment, please update the figure legend to match the additional highlighted sample

112: Add “mono” in front of “methyl-branched alkanes” for clarity

124/125: If the results in figure 2b are pooled for the B and L lineages, please include that detail in the legend. Did you check that the two lineages didn't significantly differ from each other in terms of CHC proportions? This would help justify pooling if that is the case.

Figure 3: Consider including bars/asterisks/n.s. significance indicators above each colony cycle # pair in part a of this figure.

156: Add description of median for clarity

172: Given that the authors later speculate that the reduced extranidal activity might aid in transmission, is it surprising that no transmission occurred during this experiment? Is transmission expected to occur by means not made available (or tested) in the experiment?

186: Soften this claim by adding “may.” The PG might also be attractive for other reasons, e.g., convenient infection site, ease of social transmission to other individuals, etc.

188/189: The CHC reduction could also be explained indirectly by nematode-induced changes in gene expression and/or CHC biosynthetic pathways in the ant. There are some GO terms showing up in the data that are related to lipid transport and fatty acyl-coA metabolic processes that might have effects on CHCs.

243: Change "little" to "few"

248: Change "cannot be confirmed" to "was not know." Confirmation might have been possible with genetic analyses of the source populations and the ones used in the experiments.

249/250: Include a sentence explaining why these two additional species were used in phylogenetic analyses.

313: Include a description of the tissue types.

366: Does "CHC-relevant" mean that some CHC peaks were excluded? Please explain or reword.

373: Did the authors mean 18,960? Was it really this precise? Did this happen only once at the start of the experiment, or were there other exposures throughout?

374: Add a space, i.e., 1.5g -> 1.5 g

392: Similar to 373/374: Did the authors mean 28,000? Add a space, i.e., 1.5g -> 1.5 g

404/405: Please include a brief description of how many days a colony cycle took on average (or a range of days).

Supplemental text:

Update all uses of "PPG" to "PG" as done in the main manuscript.

Table 4 legend: Add "%" to the description unless you plan to change these values to proportions out of 1.0 (instead of 100, as they currently are reported).

Reviewer #4:

Remarks to the Author:

In the present manuscript "Behavioural individuality determines infection risk in clonal ant colonies" Li and colleagues analyzed parasite infection risk in a social insect. The authors are right in their statement that experimental studies that test the link between host spatial behaviour and parasite infection risk are relatively rare. By using extensive sampling and a variety of techniques, including behavioural assays, transcriptional analyses, and chemical analyses, the authors were able to support their hypothesis that worker clonal raider ants are more likely to be infected when they leave their nests. Further, they found that infections affect the behaviour of the infected host as well as the dynamics within the whole colony (including non-infected ants), immune genes were up-regulated in infected ants and CHC profiles were affected.

The applied methods and statistical analyses are overall appropriate (see questions/comments below). The study is interesting, straight forward and adds valuable information towards our understanding of the link between individual behaviour and infection risk. It builds upon previous research on unequal infection risk of members in a population (some of which was cited) and supports these earlier studies (mainly observational in nature) with experimental data. I have several concerns/comments.

Introduction/Abstract

- The Abstract does a great job of introducing the topic and gives a rough overview of the methods used. However, neither the results of the chemical analysis, the phylogenetic analyses or the transcriptional analyses are mentioned. This should be changed.
- The very first sentence (L 28) of the Introduction needs a reference. The same goes for the sentence

spanning L 30 – 32.

- The authors do not explain how the transmission of the nematode from one host to the next happens. Please add a sentence or two. Can the nematode be horizontally transmitted? Without this kind of information, it is unclear if the change in behaviour of infected hosts has any influence on parasite transmission.

Results

- Figure 1: Please provide information on the number of images analyzed to create 1b. If it is 1, as the methods suggest, is this information robust enough to draw conclusions from it? I would appreciate a comment on that.
- L 96 -98: I recommend substituting "survival" for "survival rate" as survival rate applies to a population and survival typically to individual organisms.
- L 106 – 108: "The majority of DE genes in immune related clusters showed increased expression in infected individuals (Fig. S1, Tab. S3), consistent with an upregulation of the immune response" – Has the immune response been measured before? If so, please include a citation. If not, an increase in immune expression does not necessarily equal an upregulation of the immune response itself. Please add citations (re: immune activation) or rephrase this sentence.
- Figure 2: Please include an explanation for the abbreviation NMDS somewhere in the figure or figure description.
- Figure 3: Standard error and mean are hard to see – maybe change the color to black?

Discussion

- L 179 – 180: "Our findings support the view that some Diploscapter nematodes represent an early, incomplete transition to parasitism" – This is a very strong claim, and I am not sure based on the results of the study if this claim can be made. Your study tested the response of the ant to the parasitic form of Diploscapter nematodes and did not look at the evolution of parasitism.

Material and Methods

- The chemical assay and the phylogenetic analyses are outside of my expertise, and I am therefore unable to comment on the validity of these methods.
- L 264: Please include what was trimmed and give the parameters used for trimming (hard vs. soft trimming etc)
- Chemical Assay: Please explain how the chemicals were sampled.

Suggestions for better readability of the paper

Throughout the paper there are several places where readability could/should be improved. Following are a few suggestions:

- Some paragraphs are extremely short (3 sentences), it would aid readability to combine them.
- L 145 – 148: This reads like part of the discussion, not the results.
- Throughout the manuscript: Instead of labeling the group of ants that are exposed to the nematode as "infected", I recommend using the term "exposed". This will help the reader understand the difference between the group of exposed organisms vs actually infected ants. Alternatively, one could write "infection group" instead of "infected group".

Point by Point Response to Reviewers

General response to Reviewers' comments

We were pleased that all Reviewers found our work interesting, and we are grateful for their close reading and many constructive thoughts and suggestions. We have worked hard to address each of their comments carefully and believe that, as a consequence, our manuscript has greatly improved. Here, we provide a brief summary of how we have addressed the primary concerns, followed by our detailed responses to all comments.

All Reviewer comments are reprinted below in plain text, followed by our responses in blue. In addition to detailed responses to every comment, we also include a comprehensive record of all modifications made to the manuscript, along with the corresponding line numbers. The line numbers refer to the revised version of the manuscript with tracked changes (file "Li_Article_Revised.docx", with "All markup" and "Ballon: Show All Revisions In-line" selected or file "Li_Article_Revised.pdf") unless stated otherwise.

All Reviewers raised questions regarding **parasite transmission** (comments 1.6, 1.16, 1.21, 2.1, 3.8 and 4.3). We address them together in our general response here and additionally provide a detailed response after each relevant Reviewer's comment. The Reviewers asked whether and how horizontal transmission occurs, why we did not observe transmission in the mixed colonies in our experiment "Effects of infection on behaviour" (henceforth Experiment O1), and whether the absence of nematode transmission should alter the interpretation that changes in ant spatial distribution will facilitate transmission.

Before we conducted this study, virtually nothing was known about the transmission of *Diploscapter* in ant colonies, which we now clarify in the Introduction (L. 77: "Nematodes of this and related genera have sporadically been reported to infect ants since the late 19th century²⁵⁻³⁰, but the nature of their interaction with ants and mode of transmission remains poorly characterized"). However, our routine observation that clonal raider ants in laboratory stock colonies are not infected at eclosion but typically become infected within a few days strongly suggested that horizontal transmission occurs. To confirm this and address the questions raised by the Reviewers, **we performed two additional transmission experiments** (henceforth Experiments A1 and A2), **as well as a more detailed characterisation of the nematode life cycle** (henceforth Experiment A3).

In **Experiment A1**, we created mixed colonies of infected and uninfected ants in equal ratios, as the mixed colonies of Experiment O1. These colonies were established with either low host density (8 ants per colony, n = 10 colonies) or high host density (20 ants per colony, n = 9 colonies) in plastic containers of equal size (diameter 33 mm). 31 days later, we detected parasite transmission from the infected to the uninfected ants in 16 out of 19 colonies, with an average parasite prevalence in originally uninfected ants of $73.86 \pm 38.00\%$ (n = 19 colonies). Additionally, transmission occurred more often in the high density treatment (mean prevalence in the originally uninfected ants: $92.96 \pm 12.07\%$, n = 9 colonies) than in the low density treatment ($56.67 \pm 45.44\%$, n = 10 colonies; generalised linear mixed model, GLMM: host density, DF = 1, LR = 5.61, p = 0.02). Infection load in the newly infected ants was also higher in the high density treatment (7.32 ± 8.12 nematodes/ant, n = 80 ants) than in the low density treatment (1.63 ± 1.50 nematodes/ant, n = 19 ants; GLMM: DF = 1, LR = 6.16, p = 0.013). The reasons we observed transmission in Experiment A1 but not in the mixed colonies of Experiment O1 are likely the higher initial infection load in infected ants in Experiment A1 (mean \pm SD: 46.20 ± 22.40 nematodes/ant, n = 10) than in Experiment O1 (10.00 ± 4.16 nematodes/ant, n = 4). Additionally, the regular removal of dead ants in Experiment O1 (to facilitate automated tracking) but not in Experiment A1, may have reduced transmission in Experiment O1 if nematodes can exit ant cadavers to infect new ants. Based on these differences, we believe that although horizontal transmission would eventually have occurred in experiment O1, it would have taken longer than the duration of the experiment (between 30 and 60 days, depending on the colony) because of the low initial infection load in the infected ants and potentially the regular removal of

cadavers. To assess whether nematode transmission can be environment-mediated, 59 days after setting up the colonies of Experiment A1, we removed all remaining ants from the containers and added 6 new uninfected ants in each (now presumably contaminated) container of the high density treatment. 73 days after setting up the colonies of Experiment A1, we found infected ants in 6 out of 9 colonies, with an average prevalence in originally uninfected ants across colonies of $16.76 \pm 14.43\%$, $n = 9$ colonies. **Experiment A1 shows that horizontal transmission: 1) can readily occur under laboratory conditions**, given sufficiently high initial infection loads, **2) increases with host density**, providing indirect support for our prediction that colonies in which ants spend more time in the nest (i.e., closer to each other) should have increased transmission, and **3) can occur through the environment**. Results of Experiment A1 are now mentioned in the manuscript at L. 221-233 (Results) and L. 514-531 (Methods).

In **Experiment A2**, we assessed the ability of parasites to leave their live hosts, a possible mechanism for (both contact- and environment-mediated) horizontal transmission. To do so, we placed surface-sterilized live ants onto nematode culture plates (agar plates seeded with bacteria). We observed stereotypical nematode trails in the bacterial lawn in 4 out of 8 plates after 48 hours and a live nematode in the bacterial lawn in one plate (Video), when the host ants were still alive. This shows that **nematodes can leave live ants**. Given its limited scope, Experiment A2 is only presented in the reply letter.

Additionally, in **Experiment A3**, we imaged nematodes dissected from ant heads to show that **infecting nematodes are likely dauer larvae** (based on morphological features such as a constricted pharynx; Oleksandr Holovachov, personal communication), a developmentally arrested, non-reproductive stage, which is the typical infection stage in parasitic nematodes (Ogawa et al. 2009, Crook 2014). Because dauers do not reproduce, this indicates that *Diploscapter* nematodes do not replicate in the ant host and suggest that they may instead use hosts to obtain resources that then allow them to pursue their development outside the host. The life cycle of one *Diploscapter* isolate is now presented in a new figure panel (Figure 1c), a new supplementary figure (Supplementary Fig. 1), as well as at L. 108-111 (Results) and L. 365-386 (Methods). Experiments A2 and A3 were performed by Bhoomika Bhat, who has been included as an author.

Based on these results, we propose that ant-infecting *Diploscapter* nematodes have the following transmission mode: the nematodes infect ants as dauer larvae, obtain nutrition and/or dispersal from the host, actively leave their hosts, develop into adults and reproduce in the external environment, and the next generation of nematodes enter the dauer stage (likely upon crowding and/or starvation) and infect new ants. This would also contribute to explaining the relatively slow transmission dynamics observed in this system (see L. 215). Because this is still speculative, we only present the parts of this proposed infection cycle that are supported by data, namely the fact that nematodes found in ant heads resemble dauers, in the revised manuscript (L. 108, Figure 1c, Supplementary Figure 1).

We acknowledge that our prediction that increased host spatial overlap should facilitate parasite transmission is somewhat speculative, but hope the Reviewers will agree that the additional experiments (in particular, our finding that horizontal transmission readily occurs and that host density increases transmission) make that prediction sufficiently plausible to be mentioned in the discussion. Importantly, this prediction holds irrespective of the mode of transmission of the nematode (which we show can be at least partially environment-mediated). With purely environment-mediated horizontal transmission, increased host density in the nest should increase parasite density in the nest, which in turn should increase parasite transmission. With purely contact-based horizontal transmission (e.g. via grooming, see also comment 2.1 below), increased host density in the nest should increase physical contacts between hosts, which would also increase parasite transmission. We clarified this in our discussion (L. 280): “From the parasite’s perspective, gaining access to the nest of a social insect host is likely beneficial because, irrespective of the mode of transmission (contact-mediated vs. environment-mediated) and underlying causes of host behavioural changes (e.g. sickness behaviour, parasite manipulation), the nest is invariably the place where host density and therefore transmission opportunities are highest.”

We hope that the additional results and improved presentation will meet the Reviewers' standards.

Point-by- point response to Reviewers' Comments

Reviewer #1 (Remarks to the Author):

The authors present a fascinating study and demonstrate that behavioral individuality causes differences in infection. Most notably the authors use a tractable and unique animal system -clonal raider ants– which allows them to disentangle the effects of other factors (e.g., genotype) on infection load from behavioral variability alone. They also demonstrate that the infection can affect social organization because infected workers stay in the nest. While I find this an interesting study, I have some concerns that I would suggest being addressed before I can recommend publication.

We thank the Reviewer for these encouraging words.

General comments:

1.1 Both the title and the abstract suggest that this is a study that predominately focuses on the behavior of infected ants. But, there are parts of the manuscript which appear somewhat disconnected in its current form (and are not mentioned in the abstract/intro). For instance, the authors have a whole section on how they test for natural infection in different species, how infection changes the ants' physiology (survival and potentially fitness, including gene-expression work, CHC abundance), and where the infection is localized in the brain (Micro-CT), but the connection of these experiments to the behavioral component of the work is often a bit unclear (i.e., why were all these additional experiments included?). I, therefore, suggest clarifying this connection between the different, successive steps of the authors' experimental work. This could be achieved in several sections such as the abstract, introduction, and results (see more specific comments below).

We agree that the manuscript did not sufficiently motivate the reasons for including transcriptomic, chemical and micro-CT analyses in addition to behavioural analyses. We explain our reasons for including these data below, and now mention the morphological, chemical, and transcriptomic analyses in the abstract (L. 19) and/or introduction (L. 90-91), following the constructive suggestions of the Reviewer, for which we thank them.

When we started this study, very little was known about the association of *Diploscapter* nematodes with ants. In particular, the relationship was assumed to be phoretic (i.e., nematodes were assumed to be commensals that use ants merely to disperse (Markin and McCoy et al. 1968, Köhler 2012), but we could not find any controlled experimental studies measuring the effects of nematode infections on host fitness. Given this, we felt that a basic characterization of the ant-nematode system was needed to confirm infection localisation and assess the effects of infections on host fitness before even starting to study our core questions on how infections interact with behaviour. This was especially important as we conducted this study in the clonal raider ant, a species in which *Diploscapter* had not been reported before. Together, our first set of analyses (Figs. 1 and 2) show that *Diploscapter* naturally infects the pharyngeal gland (PG) of ants and affects the survival and physiology of their host, indicating that these nematodes are not merely phoretic, but can act as “real” parasites under certain conditions.

1.2 While the link to the data repository works, the provided DOI (<https://doi.org/10.17617/3.I6NKXM>) for the repository does not seem to exist. I suggest correcting this.

At the time of submission, we had not yet activated the public DOI link but instead provided a temporary link to the same data and R scripts to ensure that the Reviewers could access all the data:

<https://edmond.mpdl.mpg.de/privateurl.xhtml?token=8f851e3a-94a7-4e96-8225-8c080c205771>. We will activate and update the public DOI link when the manuscript is accepted for publication.

Abstract

1.3 I think the abstract addresses all the behavioral experiments clearly (and their background) but there is a whole set of experiments (e.g., PG infections, gene expression comparisons, Micro-CT, tests for natural infection) that are not mentioned which, to me, leads to a disconnect and questions of why these additional experiments were included. I think the authors could relatively easily incorporate why all these experiments were done and how they connect to the main experiment that investigated behavioral individuality and its effect on infection load (see comment above).

We agree and now mention the additional experiments in the abstract. Because the abstract is limited to 150 words, this means we had to cut other parts of the abstract, and still cannot explain each analysis and its results in detail, but think the revised abstract is more complete.

The revised abstract reads: “In social groups, infection risk is not distributed evenly across individuals. Individual behaviour is a key source of variation in infection risk, yet its effects are difficult to separate from other factors (e.g., genotype, age). Here, we combine long-term epidemiological experiments with chemical, transcriptomic, and automated behavioural analyses in clonal ant colonies, where behavioural individuality emerges among identical workers. We find that: 1) *Caenorhabditis*-related nematodes (*Diploscapter*) parasitise ant heads and affect their survival and physiology, 2) differences in infection emerge from behavioural variation alone, and reflect spatially-organised division of labour (DOL), 3) infections affect colony social organisation by causing infected workers to stay in the nest. By disproportionately infecting some workers and shifting their spatial distribution, infections reduce DOL and increase spatial overlap between hosts, which should facilitate parasite transmission. Thus, DOL, a defining feature of societies, not only shapes infection risk and distribution but is also modulated by parasites.”

Minor comment

1.4 Line 17: The authors mention other factors. I believe that this is a key sentence that distinguishes their work from other work (and highlights how unique their study system is). I would therefore suggest mentioning such other factors that could lead to variation in infection risk in the abstract.

The revised version of the abstract now mentions the two main factors that most often confound behavioural variation in social insects, namely age and genotype (L. 51, or see 1.3 above).

Introduction

1.5 Overall, the introduction is well-written. I do feel like it ends abruptly without a quick summary of what was done and what was predicted by the authors. More specifically the introduction ends with a statement about how the localization of the nematodes might affect social organization in the colony. I believe this would be an opportunity to briefly follow up connect the authors’ experimental approaches (see general comment above) by stating that they first confirmed natural infection, localization of the infection, and changes to physiology (using different methods), and then used this parasite to investigate its effects on social behaviors and downstream social organization. This statement could also include some predictions (i.e., we predicted the parasite to be localized in the PG and therefore it could affect social behaviors). I believe such a summarizing statement at the end of the introduction could clarify some of my main concerns about the disconnect of physiological and behavioral work.

We thank the Reviewer for this constructive comment. We have added a new paragraph at the end of the Introduction (L. 90-97): “Here, we first confirm that *Diploscapter* nematodes naturally infect several ant species and localise to the PG. We also combine survival, transcriptomic, and chemical

analyses of CHC profiles in experimentally infected clonal raider ants to show that these nematodes harm their host and broadly affect their physiology, indicating that they are not merely commensal. We then perform a long-term epidemiological experiment in colonies composed of identical workers to test whether workers active outside the nest face higher infection risk than otherwise identical workers active in the nest. Finally, we use experimental infections and automated tracking to test whether the presence of nematodes in the PG affects individual spatial behaviour and colony social organisation.”.

1.6 I think the introduction could benefit from a bit more details about the transmission dynamics of the nematodes (if known). Namely, (1) how do ants get infected, (2) does the parasite spread horizontally? Is increased infection load a function of replication or continued uptake from the environment? What is the life cycle, and how long from infection to shedding? These add-ons could be made in the paragraph that discusses the biology of the parasite (lines 63-75).

The reason the introduction does not say much about *Diploscapter* transmission is that virtually nothing was known on this topic before we conducted this study, which we now clarify in the Introduction (L. 79). The work on *Diploscapter* and other nematodes infecting ant heads consisted primarily of a small number of field surveys, which speculated on the nature of the relationship and the mode of transmission in the absence of controlled experiments. To fill that gap and address this and other Reviewers’ questions on transmission, we decided to perform additional experiments (see General Response above). Below we answer each of the questions of the Reviewer in light of our results from the additional experiments.

1) How do ants get infected?

Ants get infected when dauer larvae nictating in the environment attach to passing ants (Supplementary Movie 1). While transmission might in principle also occur directly (e.g., the contents of the PG are known to be transferred between ants by grooming (Soroker et al. 1995, Inwood and Morgan 2008)), we currently do not know whether and how frequently contact-mediated transmission occurs.

2) Does the parasite spread horizontally?

Yes, the nematodes spread horizontally. Our additional Experiment A1 (see General Response above) shows that horizontal transmission occurs under laboratory conditions and can be environment-mediated. In Experiment A1 (see detailed methods at L. 221-233), we created mixed colonies of infected and uninfected ants in equal ratios, as in the mixed colony treatment of the original manuscript. 31 days later, we detected parasite transmission in 16 out of 19 colonies, with an average parasite prevalence in originally uninfected ants of $73.86 \pm 38.00\%$ across colonies ($n = 19$ colonies). To assess whether nematode transmission can be environment-mediated, 59 days after setting up the colonies, we removed all remaining ants from the containers and added 6 new uninfected ants in 9 of the (now presumably contaminated) containers. 14 days later, we found infected ants in 6 out of these 9 colonies, with an average prevalence of $16.76 \pm 14.43\%$ across colonies. Experiment A1 shows that horizontal transmission can readily occur under laboratory conditions and can occur through the environment. Additionally, we know that transmission is not vertical as newly eclosed workers in infected colonies are all uninfected (as explained at L. 330). Results of Experiment A1 are now presented at L. 514-531.

3) Is increased infection load a function of replication or continued uptake from the environment?

We imaged nematodes from ant heads (Experiment A3; see also General Response above, L. 108, Fig. 1c, Supplementary Fig. 1) and show that they resemble dauer larvae, a developmentally arrested, non-reproductive stage, which is the typical infection stage of parasitic nematodes. Because dauers do not reproduce, this indicates that *Diploscapter* nematodes do not replicate in the host and suggests that they may use hosts to obtain resources that allow them to pursue their development outside the host. Therefore, increased infection load is almost certainly the result of continued uptake from the environment in our experiment (Fig. 3a, L. 181), which we now mention at L. 183.

4) What is the life cycle, and how long from infection to shedding?

We characterised the life cycle of the nematodes and present the results in Figure 1c and L. 108-111 (Results) and L. 365-386 (Methods). We currently cannot measure the time from infection to shedding because we cannot track the inflow and outflow of nematodes from hosts in real time. However, the fact that nematodes likely do not need to reproduce in the host (see point 3 above) and can exit live hosts (see Experiment A2 in General Response above) means that the time to shedding might in principle be short.

Minor comments:

1.7 Lines 37-38: I suggest mentioning these other factors more explicitly since they are important to differentiate the authors' study from other studies in this field.

We now mention some of the confounding factors at L. 51: "which are instrumental in generating hypotheses but can be confounded with other factors such as variation in age or genotype".

1.8 Line 55: Here and throughout I suggest changing to past tense. I.e., "investigated" instead of "investigate".

We now use the past tense to describe experimental procedures throughout the manuscript.

1.9 Line 58: This is a bit confusing. If they are near-identical can the authors exclude genotypic effects? Please clarify.

All individuals in clonal raider ant colonies are produced by a mode of asexual reproduction called automixis with central fusion, which results in nestmates being almost clonally related (within-colony relatedness is 0.996; Oxley et al., 2014). Relatedness is not 1, and nestmates can therefore not be said to be identical, because of (rare) mutations and losses of heterozygosity. We cannot rule out that some of this remaining genetic variation affects behaviour. However, this extremely small amount of genetic variation is very unlikely to explain the amount of behavioural variation observed among (clonal) nestmates (here and in past work, e.g., Ulrich et al. 2018), or across an individual's lifetime (Ulrich et al. 2021). The overwhelming majority of inter-individual differences in behaviour are therefore plastic, and not attributable to underlying genetic variation. We now specify what we mean by "genetically near-identical", so that the sentence now reads (L.71): "Colonies of this species are queenless and composed of genetically near-identical workers (within-colony relatedness is 0.996²²) that reproduce asexually and synchronously²³".

1.10 Line 67: I suggest either removing "Strikingly" or adding a clarification on why this specific tissue tropism is striking compared to other infection types.

We removed "strikingly" from the sentence, which now reads (L.79): "The association between nematodes and ants displays high organ-specificity: nematodes were repeatedly reported to infect a specific head gland, the pharyngeal gland (PG)²⁵⁻²⁹".

1.11 Lines 68-69: I would suggest removing the statement about the previous name of the PG. I don't think it is necessary information and might confuse the reader.

We agree and have removed the statement. The sentence now reads (L. 81): "The PG has an eminently social function in ants, with roles in the transfer of nutrients and signalling molecules between colony members^{31,32}".

Results

The results are well presented and I very much like the presented figures.

General comments:

1.12 For some of the visualizations, I wonder whether it would be worth it to make them more accessible for non-color printing. I.e., instead of using colors also use different symbols. But, I leave this up to the editors and authors discretion.

We appreciate the feedback on accessibility and addressed the comment as follows:

- Fig 2b, 2c: in addition to colour, we now use open/solid shapes to indicate infection status in the figure and figure legend (L. 160, 162).
- Fig 2d, Fig 3a: in addition to colour, we now include information on the position of data (left/right) in the figure legend (L. 164).

Both modifications will allow readers to distinguish categories without colour.

Minor comments:

1.13 Line 94: I suggest defining “phoretic”, i.e., attachment for the purpose of travel here.

We opted to simplify the language and rewrote the sentence without using the term “phoretic” (L. 126): “While previous studies proposed that *Diploscapter* nematodes are commensals that use ants merely to disperse^{28,29}, our results instead show that these nematodes harm their host, and can therefore be (facultative) larval parasites”.

1.14 Line 94: The authors state that their results indicate that the nematodes do not use their ants host for dispersal alone and that this could be driven by reduced survival of the infected workers over 51 days. From the figures, it looks like survival differences started to arise around ~10 days. Does this mean that parasites do not disperse in these early days of infection? I suggest clarification about some of the potential transmission dynamics around the presented results (e.g., in the introduction, see comments above).

This comment prompted us to clarify this sentence. What we aim to convey here is that reduced survival in nematode-infected ants strongly suggests that the nematodes are (at least facultatively) parasitic rather than merely phoretic (as previously thought). This is because phoresis is a form of commensalism which benefits the commensal (by promoting its dispersal), at no cost to the host. Reduced host survival therefore indicates that the relationship is “more than” phoretic.

We have (separately) addressed the question of transmission in this system (see General Response above and our replies to comments 1.6, 1.21, 2.1, 3.8 and 4.3), and clarified the sentence here (L.126, see also comment 1.13).

1.15 Line 115-116: I think this sentence is a great opportunity to establish the connection of why the physiological experiments were done and how they connect to the behavioral work. This could be achieved by, for instance, stating “because we also detected changes in CHC classes on the cuticle important for social interactions in this species, we next tested...”

We have added a bridging sentence which reads (L. 152): “Given the social function of the PG, we next investigated the link between nematode infection and the social organization of the colony”.

1.16 Line 138-140 This is interesting, so nurses, despite being predominantly in the nest, also get infected, and infection load increases? Does this mean there is horizontal transmission? Again, I think a description of the transmission dynamics (in the intro) could clarify some of these points.

Our additional Experiment A1 (see General Response and reply to comments 1.6, 1.21, 2.1, 3.8 and 4.3) shows that horizontal transmission indeed occurs. Increasing infection load in nurses can be explained by several mutually non-exclusive processes: (1) task specialisation in the clonal raider ant

is not strict (Ulrich et al. 2018), meaning that although nurses spend considerably more time in the nest than foragers, they do not spend *all* their time in the nest (especially across the 270 days of this experiment) and may still be exposed to nematodes outside the nest, though at a lower rate than foragers. We clarified this at L. 307, (2) nurses might become infected by nematodes brought back to the nest by foragers, either by direct contact (e.g., grooming) or via the environment (after nematodes exit foragers in the nest and reinfect new ants). While we cannot quantify the relative importance of these processes in driving colony-level infection dynamics, we believe both processes are likely at play, especially in such a long experiment.

Discussion:

General comment:

1.17 This is a general thought and might be something the authors wish to discuss. At the very end of the discussion, the authors mention that “by disrupting division of labor and increasing spatial overlap between colony members, infecting nematodes may collapse the organizational barrier to their own advantage”. But, couldn’t it also be detrimental to the parasite if the infection stays localized within the nest and it does not disperse beyond its barriers?

Thank you for this thought-provoking comment. We agree that the observed behavioural effects would intuitively be expected to increase within-colony transmission but decrease between-colony transmission. It is therefore possible that within-colony and between-colony transmission are traded-off against each other and that host behavioural changes only provide short-term transmission advantages to the parasite. However, there is no information (from either the lab or the field) on between-colony movement patterns in the clonal raider ant, making it difficult to discuss the consequences of infection-induced behavioural shifts on between-colony transmission, even speculatively. In particular, it is unclear how between-colony transmission could occur, given most ants show efficient nestmate discrimination, making it impossible for workers to “drift” between colonies in nature. Still, nematodes might be able to infect new colonies by infecting dispersing young winged queens (in sexual species) or through colony fission (the most likely colony reproduction mode in the queenless clonal raider ant). Given this is mostly speculative at this point, we chose not to elaborate on them in the manuscript. However, we would be happy to mention the potential for long-term costs in the form of reduced between-colony transmission in the last sentence of the discussion if the Reviewer thinks this is important in light of the above discussion.

Minor comments:

1.18 Lines 179-181: I am a bit confused about how the authors results indicated that this specific parasite represents an early, incomplete transition to parasitism. Please clarify.

While most authors before us have assumed a primarily phoretic relationship between *Diploscapter* and ants (i.e. a relationship that benefits the nematodes but doesn’t harm the host) (Markin and McCoy et al. 1968, Köhler 2012), some authors have proposed that the relationship might instead represent a first step towards parasitism, based on observed physical damage to the gland (Wahab 1962; see also excerpts below). By demonstrating that infections decrease survival, modulate immune activity, and affect chemistry and behaviour, our results support the claim that *Diploscapter* nematodes act as facultative larval parasites (i.e., that they harm their host).

This and another comment (4.9) prompted us to make that sentence more explicit. It now reads (L. 239): “Our findings thus support the view that some *Diploscapter* nematodes are facultative larval parasites of ants³⁵ in a clade of otherwise free-living, bacterivore nematodes related to *C. elegans*³⁹”.

Excerpts, with relevant sections underlined:

“The first indication of a step toward parasitism is found in members of the genus *Diploscapter* and

Caenorhabditis where third stage juveniles invade the pharyngeal glands of ants and in Parasitorhabditis that invades the intestine of wood-boring Coleoptera.”, Poinar, G. O. (1975) Entomogenous Nematodes: A Manual and Host List of Insect-nematode Associations. Brill, Leiden. P. 59.

“The last type of relationship that can be derived from endo-phoresy of a dauer larva is larval parasitism, where only the juvenile stage obtains nutrients or energy at the expense of the living host, irrespective of how severe the effect is on the host. [...] That this endo-phoresy was an intermediate step to larval parasitism is obvious, as some species (like *P. piniperdae* and *P. proximi*) take nutrients from the host and thus became larval-parasitic in the gut, the malpighian tubules, or even the body cavity (Rühm, 1956; Lazarevskaya in Poinar, 1972). The same was claimed for the rhabditids *Diploscapter lycostoma* and *Oscheius dolichurus* in pharyngeal glands in the heads of ants (Wahab, 1962) and similarly for *Rhabpanus ossiculum* from termites (Massey, 1971). It is not known in any of these species if the incorporation of substances from the host is essential for completion of the life cycle.” Sudhaus, W. (2008) Evolution of insect parasitism in rhabditid and diplogastrid nematodes. Advances in Arachnology and Developmental Biology, pp. 143-161.

“Species of the genus *Diploscapter* are associated with ants. The third stage juvenile nematodes of *D. lycostoma* enter the mouth of adult ants and collect in the pharyngeal glands as reported by Wahab (50). The nematodes feed in the glands, causing a reduction in the thickness of the glandular epithelium. The ants do not die and the nematodes eventually emerge and molt to the adult stage in the host environment, where they undergo continuous development. The nematodes can also be bred on cultures of raw potatoes for many generations without the presence of ants. Since Wahab noted that the nematodes increase in size within the glands, the association may be considered a parasitic one. A similar association was observed with *D. lycostoma* and laboratory reared Argentine ants, *Iridomyrmex humilis*, in California (23). Reproducing colonies of nematodes were found in refuse piles accumulated by the ants, and these piles were one of the primary sources of infection. Ants placed in contact with infested detritus were infected within 24 hours. Histological sections and direct observations indicated that the nematodes caused definite damage to the pharyngeal glands-possibly limiting their function since the gland lobes were often reduced to a fragile shell. Although the nematodes did not appear to have any decided effect on the worker ants, the secretions of these glands may play an important role in the social biology of the nest, and their destruction by nematodes could have an effect on the ant colony as a whole.” Poinar., G. O. (1972) Nematodes as Facultative Parasites of Insects. Annual Review of Entomology Vol. 17.

Note that we do not quote directly from Wahab (1962) because that work is in German.

1.19 Lines 183-185: Does immune activation always mean that the intruder derives resources from the host? Please clarify.

No. On its own, immune activation does not imply that symbionts derive resources from their host. However, together with decreased host survival, the two findings strongly suggest in our view that nematodes can derive resources from their host (i.e., can act as parasites). The original sentence was formulated to not imply causation (“However, infections induce an immune reaction and decrease host survival, indicating that these nematodes do not merely use ants for dispersal, as previously thought^{22,24,25}, but may instead derive resources from their host”). We have tried to further clarify this, so that the sentence now reads (L. 243): “Infections modulate immunity and decrease host survival, which together suggests that these nematodes do not merely use ants for dispersal, as previously thought^{28,29}, but may instead derive resources from their host”.

1.20 Lines 188-189: The statement about *C.elegans* seems disconnected. I suggest clarifying the relationship of *C.elegans* with *Diploscapter* here (sister group) to establish the connection that sterols could also affect *Diploscapter*.

We have added this information so that the sentence now reads (L. 249): “In addition, the PG of ants contains sterols^{40,41}, which *C. elegans* (which belongs to the sister taxon of *Diploscapter*) requires exogenously for survival and to develop from dauer larvae into reproductive adults⁴²”.

1.21 Lines 216-218: I think this statement could again benefit from more clarification about the transmission dynamics (see comments about the intro above). Is there horizontal transmission?

Yes, our additional experiment shows that there is horizontal transmission in this system (see also General Response and replies to comments 1.6, 1.16, 2.1, 3.8 and 4.3). We now make this explicitly clear in a new paragraph (L. 221-233).

1.22 Lines 224-225: This sentence suggests that the infected workers just continue their “work” inside the nest which I assume isn’t the case since they are probably suffering from the heavy effects of the infection. I wonder if it would make more sense to replace “workers” with “individuals” (in this sentence at least).

We agree and have replaced “workers” with “individuals” as suggested in this sentence (L. 289), which now reads: “Thus, by turning extranidal individuals into intranidal individuals, nematode infections effectively dampen division of labour at the colony level”.

Methods:

The methods are well-described and easy to follow. I also believe they provide enough detail for the work to be reproduced.

Minor comments:

1.23 Lines 240-241: The sentence read to me that the same ants tend to larvae but also leave the nest to forage during the brood-care phase, but I assume this is when reproductive division of labor occurs and some ants are nurses and others are workers, correct? Maybe slightly rewrite to clarify this?

The Reviewer is correct that division of labour arises in the brood-care phase between individuals that tend to take on tasks inside the nest (e.g. nursing) vs. outside the nest (e.g. foraging) (Ulrich et al. 2018), although in the clonal raider ant, this division of labour is not as pronounced as in species with morphologically specialised castes. In other words, individual clonal raider ants consistently vary in their tendencies to perform tasks, though no individual is solely specialized in a single task. We have clarified this, so that the section now reads (L. 305) “During the reproductive phase, all ants remain in the nest and lay eggs. During the brood-care phase, the ants attend to the growing larvae at the nest but also leave the nest to forage, explore, or dispose of waste. Individual ants vary in their tendency to perform tasks in the nest vs. outside the nest, which is reflected in their spatial behaviour²⁴”.

Reviewer #2 (Remarks to the Author):

Li and co-authors conducted a series of experiments involving various approaches that include molecular phylogeny, micro-CT, transcriptome, CHC analysis and behavioral tracking to show that behavioral variations (division of labor) by the clonal raider ant is a key determinant of infection risk posed by a parasitic nematode. They also showed that nematode infection modulates the ant’s social organization in which individuals within infected colonies overlap spatially which in turn may favor the nematode transmission.

All the experiments and analyses were well carried out. Data presentation is clear and straightforward. I particularly appreciate the micro-CT efforts that produce the very first empirical evidence for where these parasitic nematodes are located in an ant head (without dissection). Furthermore, I am happy to

see that nematodes are now added on the list for the study of interactions between sociality/social behavior and disease dynamics. In general, authors have done a terrific job addressing the key questions with solid data, and most of the conclusions are well supported by the results.

Thank you. We appreciate the Reviewer's interest in this system.

2.1 One critical question that may need a clarification is that are nematodes able to transmit between ant individuals and, if they are, what would be the primary transmission pathway and how efficient is it?

As authors mentioned, these nematodes are using nictation behavior to attach on a host ant that passes by and then enter the ant's PG. Do they have to leave the ant host's PG so they can be picked up by another ant worker? Authors must establish this baseline information to support their prediction on spatial overlapping of individuals facilitating parasite transmission. Also, authors may want to look at and test if trophallaxis serves as an effective/promising transmission route for the nematode to spread among nestmates, especially given the fact that the nematodes are localized in PG that plays some significant roles when it comes to sharing food via trophallaxis (many ants, not sure about *Ooceraea biroi* though, mix food with PG secretion before regurgitation)? If it is true that trophallaxis serves as a major way for nematodes to spread, just workers staying close with each other may not necessarily result into increase chance of transmission unless trophallaxis frequency can be established. Another potential scenario would be "sickness behavior" - reduced extranidal activity may simply be an adaptive behavioral change (e.g., ants conserve energetic resources so they are able to sustaining nematode infection) responding to nematode infection. The authors may also want to account this possibility when interpreting the results.

These are all excellent points and we conducted a series of additional experiments (see also General Response) to address them and similar comments (1.6, 1.16, 1.21, 3.8 and 4.3).

In Experiment A2, we assessed the ability of parasites to leave their live hosts, a possible mechanism for (both contact- and environment-mediated) horizontal transmission. To do so, we placed surface-sterilized live ants onto nematode culture plates (agar plates seeded with bacteria). After 48 hours, we observed stereotypical nematode trails on the bacterial lawn in 4 out of 8 plates and recorded a live nematode in one plate (Video), when the host ants were still alive. **This shows that nematodes can leave live ants.**

In Experiment A1, we show that horizontal transmission readily occurs (provided sufficient initial infection loads) and can be environment-mediated. We created 19 mixed colonies of infected and uninfected ants in equal ratios, as the mixed colonies of the original manuscript, but with a higher initial infection load (46.20 ± 22.40 nematodes/ant in Experiment A1 vs. 10.00 ± 4.16 nematodes/ant, in the original experiment). After 31 days, we detected parasite transmission in 16 out of 19 colonies, with an average prevalence in the originally uninfected ants of $73.86 \pm 38.00\%$ across colonies ($n = 19$). 59 days after setting up the colonies, we removed all remaining ants from the nest containers and added 6 new uninfected ants in 9 of the containers. 73 days after setting up the colonies, we found infected ants in 6 out of these 9 colonies, with an average prevalence of $16.76 \pm 14.43\%$ across colonies. Experiment A1 is now presented at L. 221-233 (Results) and L. 514-531 (Methods).

These experiments show that **nematodes can exit live hosts** (Experiment A2) and **can transmit horizontally**, at least partially through the environment (Experiment A1). Based on all the available evidence, we hypothesize that the environment-mediated horizontal transmission is the result of nematodes actively exiting the ants, completing a reproductive cycle in the environment and infecting new ants when new dauers are formed (see also General Response above). While we currently do not have evidence for contact-mediated transmission (nematodes are small and transparent and imaging them as they transmit between ants is very challenging), we think it is likely to occur via mutual grooming (the contents of the PG are known to be exchanged between workers by mutual grooming

(Soroker et al. 1995, Inwood and Morgan 2008)), but not by trophallaxis (clonal raider ants do not have trophallaxis between adults).

We fully agree that we cannot rule out sickness behaviour as an explanation for the decreased extranidal activity of infected individuals and now explicitly mention sickness behaviour as a possible cause for the observed behavioural changes (L. 284). However, given nematodes can actively exit their host, decreased extranidal activity due to sickness behaviour (without a corresponding increase in physical contact between ants) would still be expected to increase parasite density in the nest, and therefore, transmission opportunities for the parasite (see Experiment A1, L. 226). We now clarify in the discussion (L. 280): “From the parasite’s perspective, gaining access to the nest of a social insect host is likely beneficial because, irrespective of the mode of transmission (contact-mediated vs. environment-mediated) and underlying causes of host behavioural changes (e.g. sickness behaviour, parasite manipulation), the nest is invariably the place where host density and therefore transmission opportunities are highest”.

I also identified a few places that may use a nice justification to make the statements clearer:

2.2 L347: what are the potential confounding factors that may influence the CHC profiles of the ant during colony phase

We now outline these potential confounding factors at L. 443: “All colonies were in the reproductive phase, ruling out potentially confounding effects of variation in colony phase on CHC profiles (ants vary in behavioural, nutritional, and reproductive status across colony phases)”.

2.3 L389: I wonder if infection load was quantified in this experiment? I understand that authors did present the nematode load data in the section “Behavioural individuality determines infection risk”, but I was not sure if the nematode load data were presented in the section “Infections alter social organisation”? The reason I would like to have authors address this question is that it is likely that those infected individuals that still engaged extranidal activities be those who have a lower nematode titer (counts)?

This is a great question. We indeed measured infection load in all ants at the end of the experiment but found no statistical association between individual infection load and individual extranidal activity (Fig. R1; GLMM: extranidal activity ~ infection load + (1|colony); likelihood ratio: 0.025, $p = 0.875$). It is important to note however that there is a considerable gap between the times at which behaviour and infection load were measured. Whereas behaviour was measured over experimental days 1 to 6, infection load was measured at the end of the experiment for each colony (i.e., between days 30 and 60, depending on the colony, see also our reply to comment 3.19). Because of this time difference, the measured infection load does not necessarily reflect infection load at the time behaviour was measured, making it difficult to conclude whether there is really no relationship between infection load and behaviour, or whether we simply did not have the appropriate data to detect such an effect, if present. Given this negative result is equivocal, we only mention it in the response letter but would be open to including it in the manuscript if the Reviewer feels this is important.

Figure R1 (not shown in manuscript). Infection load at the end of the experiment and extranidal activity during the first 6 days of the experiment. Dots represent individual infected ants infected (orange) and mixed (blue) colonies.

2.4 L402-403: I appreciate the efforts that authors dissected all workers at the end of experiments. I wonder how many individuals from the infected groups actually have no nematodes in their PGs? I asked this question because this would provide readers information regarding how efficient/successful the artificial inoculation assay is (for example, does it result into 100% infection rate among workers in a group?).

Our infection protocol typically results in close to a 100% infection rate among exposed ants. This is because we check prevalence and infection load regularly after initial exposure and only start an experiment when all dissected ants are infected (e.g., 4 out of 4 ants in the experiment on the effects of infection on host behaviour, 10 out of 10 ants in the survival experiment, and 10 out of 10 ants in Experiment A1). At the end of the experiment on the effects of infection on behaviour, only 6 out of the 104 infected ants had no nematodes in their PG. While this could be due to our infection protocol resulting in a small fraction of ants not being infected, Experiment A2 shows that nematodes can leave live ants (see also General Response and response to comment 2.1), and it is therefore also possible that some infected ants “lost” all nematodes throughout the experiment. Information on the success of the infection protocol is included wherever appropriate in the manuscript (L 394, L. 492, L. 517).

2.5 Nematode movement in general: My personal experiences and also previous literature (e.g., Poinar, 2012, Psyche, 192017) suggest that these nematodes enter their host ants when the external conditions are not favorable (such as low soil humidity) and may leave the host ants when conditions are favorable again. Have the authors observed nematodes exit the ants? I assume humidity was well controlled in these experimental units, which may have created an environment with suitable conditions for the nematodes. I guess the answer to the question above (L402-403) will also address this issue.

Thank you for this insightful comment. Based on it, we conducted an additional experiment (A2) to directly test whether nematodes exit live ants (see also General Response and reply to comments 2.1 and 2.4 above). In this experiment, we placed surface-sterilized live infected ants onto nematode growth plates (agar plates with a bacterial lawn). We observed stereotypical nematode trails in the bacterial lawn in 4 out of 8 plates after 48 hours and a live nematode in the bacterial lawn in one plate (Video). This finding supports the Reviewer's suggestion that nematodes can actively exit their host

ants, in this case into a favourable external environment with high humidity and ample food. Given its limited scope, Experiment A2 is only presented in the reply letter.

2.6 The introduction section would use a nice final paragraph that summarizes what hypothesis will be tested and what tools will be used for testing these hypotheses.

We thank the Reviewer for this suggestion. This and comment 1.5 above prompted us to add a new paragraph to the introduction, which summarises the hypotheses and methodological approaches of the study. The paragraph reads (L. 90): “Here, we first confirm that *Diploscapter* nematodes naturally infect several ant species and localise to the PG. We also combine survival, transcriptomic, and chemical analyses of CHC profiles in experimentally infected clonal raider ants to show that these nematodes harm their host and broadly affect their physiology, indicating that they are not merely commensal. We then perform a long-term epidemiological experiment in colonies composed of identical workers to test whether workers active outside the nest face higher infection risk than otherwise identical workers active in the nest. Finally, we use experimental infections and automated tracking to test whether the presence of nematodes in the PG affects individual spatial behaviour and colony social organisation”.

Reviewer #3 (Remarks to the Author):

This study by Li et al. presents an exciting behavioral, chemical, and genetic investigation of nematode infection in the clonal raider ant, *Ooceraea biroi*. I find this research to be of great interest for its contribution to our understanding of the effects of *Diploscapter* infection on ant behavior (specifically, forager extranidal behavior in *O. biroi*) and well as the role of forager ant behavior in infection likelihood. The authors also present interesting results regarding the effects of nematode infection on ant gene expression – particularly as it relates to immune response and behaviorally-associated genes, as well as results on changes to *O. biroi* CHC profiles, which are important for social group cohesion and non-nestmate discrimination, among other roles. In general, the authors do an excellent job of summarizing their findings and discussing the implications of their work and I think this study opens a lot of interesting questions about what might be happening behaviorally inside the colony during infection, how transmission occurs, and whether ants might be able to detect infected individuals through changes in CHCs. While most of my detailed comments in the manuscript relate to minor suggested edits and/or clarifications, a few comments relate to some of the conclusions drawn (summarized below). I have also uploaded my detailed, in-text comments included as attachments for both the manuscript and supplemental figures.

We thank the Reviewer for these encouraging words.

Comments/suggestions (by line number):

Main manuscript:

3.1 59: Missing a comma after i.e.

The comma was added (L. 74).

3.2 81: Highlight the *P. advenus* sample in Figure 1a to make it easier for readers to identify

The *P. advenus* sample is now highlighted in Figure 1a. We also adjusted colours for isolates from *L. niger* and *P. longicornis* for better readability.

3.3 90: Relating to the previous comment, please update the figure legend to match the additional highlighted sample

The legend of Figure 1a. has been updated and now reads (L. 115): “(a) *Diploscapter* isolated from the heads of clonal raider ants (green) and two other ant species (brown) are phylogenetically nested within the *Diploscapter* clade (sequence accession numbers in brackets), which includes *D. formicidae* isolated from ants³⁰ (yellow). Node labels indicate branch support (%) from 1000 bootstrap replicates. Scale bar: mean number of 0.02 substitutions per site.”.

3.4 112: Add “mono” in front of “methyl-branched alkanes” for clarity

We have changed “methyl-branched alkanes” to “monomethyl-branched alkanes” throughout the manuscript.

3.5 124/125: If the results in figure 2b are pooled for the B and L lineages, please include that detail in the legend. Did you check that the two lineages didn’t significantly differ from each other in terms of CHC proportions? This would help justify pooling if that is the case.

We thank the Reviewer for this suggestion. Yes, our statistical analysis shows that lineages B and M* do not differ in the relative abundance of either of the three classes of CHCs (Mann–Whitney U tests, pooled infection status: n-alkane: $W = 216$, $p = 0.678$; monomethyl-branched alkanes: $W = 260$, $p\text{-value} = 0.108$; dimethyl-branched alkanes, $W = 141$, $p\text{-value} = 0.114$), which justifies pooling the genotypes in Fig. 2b.

However, this comment prompted us to adjust the statistical analyses for clarification. Instead of using Mann-Whitney U tests with samples pooled across genotypes, we now use linear mixed models with infection status (infected vs. uninfected) as a fixed effect and genotype as a random effect to model the relative abundance of different CHC classes (see methods at L. 545). The corresponding section of the results (L. 147) now read “both n-alkanes and monomethyl-branched alkanes had lower relative abundance in infected individuals (linear mixed models (LMM), infection status, n-alkanes: $DF = 1$, $LR = 47.67$, $p < 0.0001$; monomethyl-branched alkanes: $DF = 1$, $LR = 15.20$, $p < 0.0001$), while dimethyl-branched alkanes had greater relative abundance ($DF = 1$, $LR = 37.18$, $p < 0.0001$)”.

*Note that this lineage was incorrectly labelled as L in the original manuscript; we have now corrected it to M throughout the manuscript to maintain consistent naming across studies and research groups

3.6 Figure 3: Consider including bars/asterisks/n.s. significance indicators above each colony cycle # pair in part a of this figure.

Bars and symbols for statistical significance were added to Fig. 3a. The legend now reads (L. 196): “Statistical significance symbols represent differences in prevalence for colony cycle 1 and differences in infection load for colony cycles 2 – 4”.

3.7 156: Add description of median for clarity

We added a description and the sentence (L.199) now reads “Thick black bars indicate the interquartile range around the median (white circle)...”.

3.8 172: Given that the authors later speculate that the reduced extranidal activity might aid in transmission, is it surprising that no transmission occurred during this experiment? Is transmission expected to occur by means not made available (or tested) in the experiment?

It was indeed surprising that no transmission occurred in the mixed colonies of that experiment (Experiment O1). This and related comments (1.6, 1.16, 1.21, 2.1 and 4.3) prompted us to perform additional experiments on nematode transmission (Experiments A1-3, see General Response above). In Experiment A1, we created 19 mixed colonies of infected and uninfected ants in equal ratios, as the mixed colonies in Experiment O1. After 31 days, we detected parasite transmission in 16 out of 19

colonies, with an average transmission efficiency (i.e., the prevalence in originally uninfected ants) of $73.86 \pm 38.00\%$ ($n = 19$) across colonies. This shows that transmission can readily occur in laboratory conditions and can be efficient. The reasons we observed transmission in Experiment A1 but not in the mixed colonies in the original manuscript (Experiment O1) are the higher initial infection loads in the infected ants in Experiment A1 (mean \pm SD: 46.20 ± 22.40 nematodes/ant, $n = 10$) than in experiment O1 (10.00 ± 4.16 nematodes/ant, $n = 4$) and perhaps also the regular removal of cadavers (to facilitate automated tracking) in experiment O1 but not in Experiment A1. Based on these results, we believe that transmission would eventually have occurred in experiment O1, but would have taken longer than the duration of the experiment (between 30 and 60 days, depending on the colony) because of the low initial infection loads in the infected ants and potentially the regular removal of cadavers. Experiment A1 is now presented at L. 221-233 (Results) and L. 514-531 (Methods).

3.9 186: Soften this claim by adding “may.” The PG might also be attractive for other reasons, e.g., convenient infection site, ease of social transmission to other individuals, etc.

We agree with the Reviewer. The sentence now reads (L. 246): “We suggest that the contents of the PG may make it an attractive source of nutrients for nematodes”.

3.10 188/189: The CHC reduction could also be explained indirectly by nematode-induced changes in gene expression and/or CHC biosynthetic pathways in the ant. There are some GO terms showing up in the data that are related to lipid transport and fatty acyl-coA metabolic processes that might have effects on CHCs.

The Reviewer makes an interesting observation and point. We agree that infections could in principle directly affect CHCs biosynthesis by the host. However, we think this is not the most likely explanation for changes in host CHC profiles, for two reasons: 1) RNAseq was performed on PGs, and (to the best of our knowledge) CHCs are not synthesized in the PG (the gland only stores and spreads CHCs), but in specialized cells (oenocytes) that are embedded in the epidermis or dispersed within the fat body. Therefore, differential expression of genes linked to lipid metabolism in the PG does likely not reflect changes in CHC biosynthesis. However, 2) the PG itself is involved in lipid metabolism, including acyl-coA dehydrogenase activity (Decio et al. 2016). It is therefore more likely in our view that infection-induced changes in gene expression reflect changes in the function of the gland itself following infection with nematodes, rather than changes in the CHC biosynthetic pathway in other parts of the host body.

3.11 243: Change “little” to “few”

We have implemented the change (L. 310).

3.12 248: Change “cannot be confirmed” to “was not know.” Confirmation might have been possible with genetic analyses of the source populations and the ones used in the experiments.

We have incorporated the suggestion (L. 315).

3.13 249/250: Include a sentence explaining why these two additional species were used in phylogenetic analyses.

The species were selected based on availability alone. We added an explanation at L. 316: “To examine whether *Diploscapter* nematodes naturally infect other ants from different geographic locations, we dissected workers from other ant species that were available at the time of the study and obtained two further nematode isolates from workers of *Paratrechina longicornis* originally collected in Malaga, Spain, and *Lasius niger* workers collected in Lausanne, Switzerland”.

3.14 313: Include a description of the tissue types.

We added information on the tissue type. The sentence now reads (L. 409): “Raw sequencing reads for infected and uninfected brains and PGs have been deposited in NCBI’s sequence read archive under the following BioProject accession: PRJNA791185”.

3.15 366: Does “CHC-relevant” mean that some CHC peaks were excluded? Please explain or reword.

The original sentence was meant to convey that some non-CHC peaks (e.g., contaminations) were removed. We have removed “relevant” from the sentence for clarity so that it now reads (L.459): “To calculate the relative abundance of CHC compounds, the area under each hydrocarbon peak in the chromatogram was quantified through integration and divided by the total area under all hydrocarbon peaks.”.

3.16 373: Did the authors mean 18,960? Was it really this precise? Did this happen only once at the start of the experiment, or were there other exposures throughout?

We thank the Reviewer for pointing out this mistake. Yes, we meant 18,960 in the original text. This number was calculated by multiplying the volume (100 μ L) and concentration of the nematode inoculum (189.6 nematodes/ μ L, obtained by averaging three independent counts). We only inoculated the agar once at the beginning of the experiment. We have edited the sentence to improve clarity. It now reads (L. 469): “For each infected colony, 1.5 g of agar was placed in the box and seeded once with 100 μ L of a nematode inoculum (concentration: 189.60 nematodes/ μ L)”.

3.17 374: Add a space, i.e., 1.5g -> 1.5 g

We made the suggested change (L. 469).

3.18 392: Similar to 373/374: Did the authors mean 28,000? Add a space, i.e., 1.5g -> 1.5 g

Yes. We rewrote the sentence (L. 490) to “... while ants from the infected group were exposed to 100 μ L of a nematode inoculum (concentration: 280 nematodes/ μ L) seeded onto 1.5 g of agar”.

3.19 404/405: Please include a brief description of how many days a colony cycle took on average (or a range of days).

We added information on cycle duration (L. 505): “For each colony, the experiment ended when all larvae had eclosed into new adults (i.e., when the colony completed a cycle, which took on average 33.96 ± 7.06 days, range: 30 – 54 days) or after 60 days (if the colony failed to complete a cycle)”.

Supplemental text:

3.20 Update all uses of “PPG” to “PG” as done in the main manuscript.

All uses of “PPG” have been updated to “PG” in the supplementary materials.

3.21 Table 4 legend: Add “%” to the description unless you plan to change these values to proportions out of 1.0 (instead of 100, as they currently are reported).

“%” has been added to the legend, which now reads (L. 39 in Supplementary Materials) “Median relative abundance \pm SD (%) of all CHCs in ants with different genotypes (B, M) and infection status (uninfected, infected).”.

Reviewer #4 (Remarks to the Author):

In the present manuscript “Behavioural individuality determines infection risk in clonal ant colonies” Li and colleagues analyzed parasite infection risk in a social insect. The authors are right in their

statement that experimental studies that test the link between host spatial behaviour and parasite infection risk are relatively rare. By using extensive sampling and a variety of techniques, including behavioural assays, transcriptional analyses, and chemical analyses, the authors were able to support their hypothesis that worker clonal raider ants are more likely to be infected when they leave their nests. Further, they found that infections affect the behaviour of the infected host as well as the dynamics within the whole colony (including non-infected ants), immune genes were up-regulated in infected ants and CHC profiles were affected.

The applied methods and statistical analyses are overall appropriate (see questions/comments below). The study is interesting, straight forward and adds valuable information towards our understanding of the link between individual behaviour and infection risk. It builds upon previous research on unequal infection risk of members in a population (some of which was cited) and supports these earlier studies (mainly observational in nature) with experimental data. I have several concerns/comments.

We appreciate the Reviewer's interest in the study.

Introduction/Abstract

4.1 • The Abstract does a great job of introducing the topic and gives a rough overview of the methods used. However, neither the results of the chemical analysis, the phylogenetic analyses or the transcriptional analyses are mentioned. This should be changed.

We agree with this and similar comments (see 1.1 and 1.3 above) and now mention the chemical and transcriptomic analyses in the abstract (L. 19) and/or introduction (L. 91). Because the abstract is limited to 150 words, this means we had to cut other parts of the abstract, and still cannot explain each analysis and its results in detail, but think the revised abstract is more complete.

The revised abstract now reads: "In social groups, infection risk is not distributed evenly across individuals. Individual behaviour is a key source of variation in infection risk, yet its effects are difficult to separate from other factors (e.g., genotype, age). Here, we combine long-term epidemiological experiments with chemical, transcriptomic, and automated behavioural analyses in clonal ant colonies, where behavioural individuality emerges among identical workers. We find that: 1) *Caenorhabditis*-related nematodes (*Diploscapter*) parasitise ant heads and affect their survival and physiology, 2) differences in infection emerge from behavioural variation alone, and reflect spatially-organised division of labour (DOL), 3) infections affect colony social organisation by causing infected workers to stay in the nest. By disproportionately infecting some workers and shifting their spatial distribution, infections reduce DOL and increase spatial overlap between hosts, which should facilitate parasite transmission. Thus, DOL, a defining feature of societies, not only shapes infection risk and distribution but is also modulated by parasites".

4.2 • The very first sentence (L 28) of the Introduction needs a reference. The same goes for the sentence spanning L 30 – 32.

We have added the following references to support our statements:

- In the first sentence of the introduction (originally L. 28, now L. 41) "Host spatial behaviour is a key driver of parasite infection and transmission success", we cite Altizer, S., Bartel, R. & Han, B. A. Animal Migration and Infectious Disease Risk. *Science* 331, 296–302 (2011).) and Allen, T. et al. Global hotspots and correlates of emerging zoonotic diseases. *Nat. Commun.* 8, 1124 (2017).
- In the sentence (originally L. 30-32, now L. 43) "Infections can in turn have drastic effects on host spatial behaviour, ranging from manipulations that increase parasite transmission to new hosts to host behavioural responses that reduce transmission.", we now cite Hughes, D. and Libersat, H. Parasite manipulation of host behavior. *Current Biology* 29, (2019): R45-R47. and Stockmaier, S. et al. Infectious diseases and social distancing in nature. *Science* 371, eabc8881 (2021)

4.3 • The authors do not explain how the transmission of the nematode from one host to the next happens. Please add a sentence or two. Can the nematode be horizontally transmitted? Without this kind of information, it is unclear if the change in behaviour of infected hosts has any influence on parasite transmission.

This important point was also mentioned by other Reviewers (see comments 1.6, 1.16, 1.21, 2.1 and 3.8, and General Response above).

We did not mention transmission in our introduction because virtually nothing was known about the transmission of *Diploscapter* in ant colonies before we conducted this study. To fill this gap and address Reviewers' questions on transmission, we performed a series of additional experiments (see also General Response). In particular, Experiment A1 shows that horizontal transmission can readily occur under laboratory conditions and is more efficient when host density is high. In Experiment A1, we created 19 mixed colonies of infected and uninfected ants in equal ratios, as the mixed colonies of the original manuscript (Experiment O1). These colonies were established with either low host density (8 ants per colony, n = 10 colonies) or high host density (20 ants per colony, n = 9 colonies). After 31 days, we detected parasite transmission in 16 out of 19 colonies, with an average transmission efficiency (i.e., the prevalence in the originally uninfected ants) of $73.86 \pm 38.00\%$ (n = 19 colonies) across colonies. Additionally, transmission was higher when host density was high ($92.96 \pm 12.07\%$, n = 9 colonies) than when host density was low ($56.67 \pm 45.44\%$, n = 10 colonies; GLMM: host density, DF = 1, LR = 5.61, p = 0.02). The finding that host density increases parasite transmission provides indirect support for our prediction that colonies in which ants spend more time in the nest (i.e., closer to each other) as a result of infection should have increased transmission. Experiment A1 is presented in the manuscript at L. 221-233 (Results) and L. 514-531 (Methods).

We acknowledge that our prediction that increased host spatial overlap should facilitate parasite transmission is still somewhat speculative, but hope the Reviewer will agree that our finding that horizontal transmission readily occurs and that host density increases transmission makes that prediction sufficiently plausible to be mentioned in the discussion. Importantly, this prediction holds irrespective of the mode of transmission of the nematode. With purely environment-mediated horizontal transmission, increased host density in the nest should increase parasite density in the nest (because nematodes exit live ants, see Experiment A2), which in turn should increase parasite transmission. With purely contact-based horizontal transmission (e.g. via grooming, see also 2.1 above), increased host density in the nest should increase physical contact between hosts, which would also increase parasite transmission. We clarified this in the discussion (L. 280).

Results

4.4 • Figure 1: Please provide information on the number of images analyzed to create 1b. If it is 1, as the methods suggest, is this information robust enough to draw conclusions from it? I would appreciate a comment on that.

We now specify in the legend of Fig. 1B (L. 119) that one specimen per host species was used to produce the micro-CT images: "Micro-CT reconstructions confirm that nematodes (red) localise to the PG (purple) in all studied ant species (one specimen per species)". This is standard, given producing these images is time- and labour-intensive. The conclusion that nematodes infect the PG of ants is not only based on the micro-CT scans but also on hundreds of dissections. The micro-CT scans therefore confirm and visually illustrate the localisation of infections to the PG, revealing details on the relative size and arrangement of the gland across ant species, and the position and density of nematodes within the gland.

4.5 • L 96 -98: I recommend substituting "survival" for "survival rate" as survival rate applies to a population and survival typically to individual organisms.

We have made the suggested change throughout the manuscript.

4.6 • L 106 – 108: “The majority of DE genes in immune related clusters showed increased expression in infected individuals (Fig. S1, Tab. S3), consistent with an upregulation of the immune response” – Has the immune response been measured before? If so, please include a citation. If not, an increase in immune expression does not necessarily equal an upregulation of the immune response itself. Please add citations (re: immune activation) or rephrase this sentence.

This is a very good point. No, the immune response has not yet been measured functionally using standard immune assays (e.g., bacterial growth inhibition, phenoloxidase) and we therefore cannot equate upregulated immune gene expression with upregulated immune activity. We have rephrased this sentence so that it now reads (L. 140): “The majority of DE genes in immune-related clusters showed increased expression in infected individuals (Supplementary Fig.2, Supplementary Table 3), consistent with a modulation of the immune response”.

4.7 • Figure 2: Please include an explanation for the abbreviation NMDS somewhere in the figure or figure description.

We included the explanation and the sentence now reads (L. 159) “Gene expression in the brain (circles) and PG (triangles) of infected (orange, open symbols) and uninfected (green, solid symbols) ants, visualised with a nonmetric multidimensional scaling (NMDS) analysis”.

4.8 • Figure 3: Standard error and mean are hard to see – maybe change the color to black?

We now added horizontal bars to the standard errors in Fig. 3a to improve readability. We also changed the colour of the standard error bar and mean in Fig. 3b (and Fig. 2d) to black as suggested.

Discussion

4.9 • L 179 – 180: “Our findings support the view that some *Diploscapter* nematodes represent an early, incomplete transition to parasitism” – This is a very strong claim, and I am not sure based on the results of the study if this claim can be made. Your study tested the response of the ant to the parasitic form of *Diploscapter* nematodes and did not look at the evolution of parasitism.

This and a similar Reviewer comment (1.18) prompted us to rephrase that sentence, which now reads (L. 239): “Our findings thus support the view that some *Diploscapter* nematodes are facultative larval parasites of ants³⁵ in a clade of otherwise free-living, bacterivore nematodes related to *C. elegans*³⁹”.

Material and Methods

The chemical assay and the phylogenetic analyses are outside of my expertise, and I am therefore unable to comment on the validity of these methods.

4.10 • L 264: Please include what was trimmed and give the parameters used for trimming (hard vs. soft trimming etc)

We now included this information at L. 336: “We Sanger sequenced the SSU (18S rDNA) of three nematode isolates following Holterman et al.⁵⁵. Sanger sequences were trimmed with Geneious Prime (v. 2021.0.3) to remove low quality bases using default options. Trimmed sequences were then assembled using the assembly tool in Geneious (with default options) to produce a consensus sequence”.

4.11 • Chemical Assay: Please explain how the chemicals were sampled.

The chemicals were extracted from pooled ant samples, which is explained in the original version of the manuscript at L. 441 (“To quantify differences in CHC profiles between infected and uninfected ants, we sampled ants from infected and uninfected colonies of two genotypes (B and M; 50 workers for each genotype and infection status combination)”) and L. 449 (“CHCs were extracted from pooled

samples by immersion in 1 mL of hexane for 10 min. Extracts were then evaporated to a volume of approximately 15 μ L of which 1 μ L was analyzed using a 6890 gas chromatograph (GC) coupled to a 5975 mass selective detector (MS) (Agilent Technologies, Waldbronn, Germany”).

Suggestions for better readability of the paper

Throughout the paper there are several places where readability could/should be improved. Following are a few suggestions:

4.12 • Some paragraphs are extremely short (3 sentences), it would aid readability to combine them.

We now have combined paragraphs 3 and 4 in the introduction.

4.13 • L 145 – 148: This reads like part of the discussion, not the results.

We have removed the second part of the sentence (“showing that differences in infection can emerge from differences in behaviour alone”) so that it now simply reads (L. 188): “Thus, the distribution of parasites across hosts reflected inter-individual variation in host spatial behaviour driven by age- and genotype-independent division of labour”. We hope the Reviewer will agree this sentence simply summarizes, but does not discuss, the results presented in this section.

4.14 • Throughout the manuscript: Instead of labeling the group of ants that are exposed to the nematode as “infected”, I recommend using the term “exposed”. This will help the reader understand the difference between the group of exposed organisms vs actually infected ants. Alternatively, one could write “infection group” instead of “infected group”.

While we agree this is an important distinction, we always checked prevalence and infection load regularly after initial exposure and only started each experiment presented in this study when all dissected ants were infected (e.g., 4 out of 4 ants in the experiment on the effects of infection on host behaviour, 10 out of 10 ants in the survival experiment, 10 out of 10 ants in Experiment A1, see also the reply to comment 2.4), ensuring that our exposure protocol actually resulted in infection. We therefore hope the Reviewer agrees that it is appropriate to use “infected” here. If not, we would be happy to change to “infection group” and “no-infection group”, if the latter is deemed appropriate.

References

- Crook, M. The Dauer Hypothesis and the Evolution of Parasitism: 20 years on and Still Going Strong. *International Journal for Parasitology* 2014, 44 (1), 1–8. <https://doi.org/10.1016/j.ijpara.2013.08.004>.
- Decio, P.; Vieira, A. S.; Dias, N. B.; Palma, M. S.; Bueno, O. C. The Postpharyngeal Gland: Specialized Organ for Lipid Nutrition in Leaf-Cutting Ants. *PLoS One* 2016, 11 (5), e0154891. <https://doi.org/10.1371/journal.pone.0154891>.
- Inwood, M.; Morgan, P. Chemical Sorcery for Sociality: Exocrine Secretions of Ants (Hymenoptera: Formicidae). *Myrmecological News* 2008, 11, 79–90.
- Köhler, A. Nematodes in the Heads of Ants Associated with Sap Flux and Rotten Wood. *Nematology* 2012, 14 (2), 191–198. <https://doi.org/10.1163/138855411x584142>.
- Markin, G. P.; McCoy, C. W. The Occurrence of a Nematode, *Diploscapter Lycostoma*, in the Pharyngeal Glands of the Argentine Ant, *Iridomyrmex Humilis*. *Annals of the Entomological Society of America* 1968, 61 (2), 505–509. <https://doi.org/10.1093/aesa/61.2.505>.
- Ogawa, A.; Streit, A.; Antebi, A.; Sommer, R. J. A Conserved Endocrine Mechanism Controls the Formation of Dauer and Infective Larvae in Nematodes. *Current Biology* 2009, 19 (1), 67–71. <https://doi.org/10.1016/j.cub.2008.11.063>.
- Oxley, P. R.; Ji, L.; Fetter-Pruneda, I.; McKenzie, S. K.; Li, C.; Hu, H.; Zhang, G.; Kronauer, D. J. C. The Genome of the Clonal Raider Ant *Cerapachys Biroi*. *Current Biology* 2014, 24 (4), 451–458. <https://doi.org/10.1016/j.cub.2014.01.018>.
- Soroker, V.; Vienne, C.; Hefetz, A. Hydrocarbon Dynamics within and between Nestmates in *Cataglyphis Niger* (Hymenoptera: Formicidae). *Journal of Chemical Ecology* 1995, 21 (3), 365–378. <https://doi.org/10.1007/BF02036724>.
- Ulrich, Y.; Saragosti, J.; Tokita, C. K.; Tarnita, C. E.; Kronauer, D. J. C. Fitness Benefits and Emergent Division of Labour at the Onset of Group Living. *Nature* 2018, 560 (7720), 635–638. <https://doi.org/10.1038/s41586-018-0422-6>.
- Ulrich, Y.; Kawakatsu, M.; Tokita, C. K.; Saragosti, J.; Chandra, V.; Tarnita, C. E.; Kronauer, D. J. C. Response Thresholds Alone Cannot Explain Empirical Patterns of Division of Labor in Social Insects. *PLoS Biology* 2021, 19 (6), e3001269. <https://doi.org/10.1371/journal.pbio.3001269>.
- Wahab, A. Untersuchungen Über Nematoden in Den Drüsen Des Kopfes Der Ameisen (Formicidae). *Zeitschrift für Morphologie und Ökologie der Tiere* 1962, 52 (1), 33–92. <https://doi.org/10.1007/bf00446341>.

Reviewers' Comments:

Reviewer #1:

Remarks to the Author:

I believe the authors have done a great job addressing comments and concerns and even included additional experiments to solidify some of their conclusions. The manuscript itself is now an impressive compilation of work including detection of natural infections, physiological work (survival and fitness, gene-expression work, CHS abundance, Micro-CT), and behavioral work describing the effects of *Diploscapter* on clonal raider ants.

Minor comment

The only small point I would like to raise is that I believe that the authors experimental demonstration that nematodes can leave live ants is an important detail and contributes to clarifying several reviewer comments/concerns. I therefore encourage the authors to incorporate Experiment A2 into the manuscript(the authors mention that they only present it in the reply letter).

Reviewer #3:

Remarks to the Author:

Overall, I am happy with the revisions. The authors responded thoroughly to the remarks and suggestions made by myself and other reviewers and satisfactorily addressed many edits and clarifications throughout the manuscript. They also addressed my concerns regarding the results presented in Figure 2b and provided additional statistical support using Mann Whitney U tests and LMM, which better justified the pooling of lineages. The authors made improvements to the figures and figure legends as requested, and incorporated suggestions for clearer explanations and wording across multiple sections of the manuscript. Lastly, the authors performed additional experiments on nematode transmission, addressing several questions and comments made by myself and other reviewers about the mode of transmission in the original experiment. I appreciate the authors' efforts and their responsiveness to feedback demonstrates dedication to improving the manuscript's quality and achieving scientific rigor. I am, therefore, recommending acceptance of the publication.

Point by Point Response to Reviewers **Final Revision**

REVIEWERS' COMMENTS

Reviewer #1 (Remarks to the Author):

I believe the authors have done a great job addressing comments and concerns and even included additional experiments to solidify some of their conclusions. The manuscript itself is now an impressive compilation of work including detection of natural infections, physiological work (survival and fitness, gene-expression work, CHS abundance, Micro-CT), and behavioral work describing the effects of *Diploscapter* on clonal raider ants.

Minor comment

The only small point I would like to raise is that I believe that the authors experimental demonstration that nematodes can leave live ants is an important detail and contributes to clarifying several reviewer comments/concerns. I therefore encourage the authors to incorporate Experiment A2 into the manuscript (the authors mention that they only present it in the reply letter).

We incorporated Experiment A2 in our manuscript at L. 187 (Results) and L.47 (Methods).

Reviewer #3 (Remarks to the Author):

Overall, I am happy with the revisions. The authors responded thoroughly to the remarks and suggestions made by myself and other reviewers and satisfactorily addressed many edits and clarifications throughout the manuscript. They also addressed my concerns regarding the results presented in Figure 2b and provided additional statistical support using Mann Whitney U tests and LMM, which better justified the pooling of lineages. The authors made improvements to the figures and figure legends as requested, and incorporated suggestions for clearer explanations and wording across multiple sections of the manuscript. Lastly, the authors performed additional experiments on nematode transmission, addressing several questions and comments made by myself and other reviewers about the mode of transmission in the original experiment. I appreciate the authors' efforts and their responsiveness to feedback demonstrates dedication to improving the manuscript's quality and achieving scientific rigor. I am, therefore, recommending acceptance of the publication.